# Mathematical biases in the calculation of the Living Planet Index lead to overestimation of vertebrate population decline

Anna Toszogyova [1] ✉, Jan Smyčka [1] & David Storch [1,2]

The Living Planet Index (LPI) measures the average change in population size of vertebrate species over recent decades and has been repeatedly used to assess the changing state of nature. The LPI indicates that vertebrate populations have decreased by almost 70% over the last 50 years. This is in striking contrast with current studies based on the same population time series data that show that increasing and decreasing populations are balanced on average. Here, we examine the methodological pipeline of calculating the LPI to search for the source of this discrepancy. We find that the calculation of the LPI is biased by several mathematical issues which impose an imbalance between detected increasing and decreasing trends and overestimate population declines. Rather than indicating that vertebrate populations do not substantially change, our findings imply that we need better measures for providing a balanced picture of current biodiversity changes. We also show some modifications to improve the reliability of the LPI.

The indicators of biodiversity change are of paramount importance for monitoring and understanding current biodiversity crisis. An influential methodology has been provided by the World Wildlife Fund (WWF) in collaboration with the World Conservation Monitoring Centre in 1997, and is commonly known as the Living Planet Index (LPI)[1]. The LPI uses available population time series to calculate the average trend in populations of vertebrate species from terrestrial, freshwater, and marine ecosystems[1–3]. It was first published in the WWF's Living Planet Report 1998[4], and in a collaborative partnership with the Zoological Society of London has been reported every two years. In 2006 it was adopted by the Convention on Biological Diversity (CBD) as one of the headline indicators of progress towards its Strategic Plan for Biodiversity 2011-2020[5] with its Aichi targets. Now it is a component indicator of the Kunming-Montreal Global Biodiversity Framework[6]. The LPI has also been adopted by the Intergovernmental Science-Policy Platform on Biodiversity and Ecosystem Services (IPBES). The LPI message is often reported in media and has become a key tool for convincing the public that the changing state of nature is serious and requires solutions. The most recently published Living Planet Report 2022[7] shows an average 69% decrease in almost 32,000 monitored

populations of mammals, birds, amphibians, reptiles, and fish between 1970 and 2018, although there is variation among biogeographical regions and ecosystem types[7].

The worrying overall decline of vertebrate populations indicated by the LPI contrasts with several current studies based on the same type of data that show that population increases and decreases are surprisingly well balanced[8,9]. Moreover, the removal of less than 3% of the most declining vertebrate populations completely reverses the overall population trend as expressed by the LPI towards an overall increase, revealing a strong sensitivity of the LPI to extreme population trends[10]. These findings have raised the question whether there is not a bias in the calculation of the LPI. One such bias may stem from the weighted averaging procedure, when the taxa and regions are weighted by the estimated species richness of respective groups. The weighted form of the global LPI shows a decline which is 38% greater than the unweighted form[3] (see also Fig. 1, Supplementary Table 1 and Methods 'Calculating the Living Planet Index'). The weighting is not necessarily a problem per se, but weighting by the species richness of a given taxon and region means that the poorly represented species-rich regions (typically tropical ones) may be driving the global LPI

[1]Center for Theoretical Study, Charles University & Czech Academy of Sciences, CZ-110 00, Prague, Czech Republic. [2]Department of Ecology, Faculty of Science, Charles University, CZ-128 44, Prague, Czech Republic. ✉e-mail: toszogyova@cts.cuni.cz

|  | Original | ts >= 5 records | ts >= 5 years | no zeros | no zeros ts >= 5 records | no zeros ts >= 5 years |
|---|---|---|---|---|---|---|
| **Global** | 0.327 | 0.474 | 0.310 | 0.519 | 0.639 | 0.552 |
| **Terrestrial** | 0.368 | 0.423 | 0.377 | 0.706 | 0.965 | 0.704 |
| **Freshwater** | 0.181 | 0.322 | 0.121 | 0.373 | 0.412 | 0.347 |
| **Marine** | 0.526 | 0.782 | 0.650 | 0.532 | 0.657 | 0.689 |

|  | no weights | no weights ts >= 5 records | no weights ts >= 5 years | no weights no zeros | no weights no zeros ts >= 5 records | no weights no zeros ts >= 5 years |
|---|---|---|---|---|---|---|
| **Global** | 0.772 | 1.004 | 0.743 | 0.989 | 1.155 | 0.961 |
| **Terrestrial** | 0.516 | 0.613 | 0.525 | 0.820 | 1.075 | 0.778 |
| **Freshwater** | 0.654 | 0.891 | 0.514 | 0.910 | 0.975 | 0.864 |
| **Marine** | 1.361 | 2.012 | 1.522 | 1.299 | 1.579 | 1.334 |

**Fig. 1 | The LPI resulting from various settings.** The calculation of the global LPI and the LPI for each ecosystem adjusted (i) by increasing the number of records in individual populations included (time series with at least 5 records), (ii) by increasing the length of the population series included (time series at least 5 years long), (iii) by removing zeros from the population time series, (iv) by removing zeros from the population time series and including those with at least 5 records, (v) by removing zeros from the population time series and including those at least 5 years long, (vi) by not using the weights (compensating different species richness) for taxa and realms, (vii) and the unweighted LPI of the time series with at least 5 records, or (viii) at least 5 years long. (ix) The unweighted LPI without zeros in the population time series, (x) and unweighted without zeros in the population time series included with at least 5 records, or (xi) at least 5 years long. The values represent the final LPI values (the value of 1 was set for 1970). The green gradient shows the rate of decrease in the index decline (a positive difference between the adjusted and original value−the adjusted index declines less than the original). The red gradient shows the rate of increase in the index decline (a negative difference between the adjusted and original value−the adjusted index declines more than the original). For an extended version of the table, see Supplementary Table 1.

trajectory (see also ref. 11). Another potential issue is that the data used for the LPI calculation include many extremely short time series, which are prone to high measurement errors due to interannual variability and sampling issues[12–14].

Recently, Buschke et al.[15] pointed out several other potential sources of bias in the LPI calculation. One problem can be due to using the GAM method for smoothing the time series and the fact that LPI values are affected by the values of the previous period (see Fig. 2). A general feature of GAM models is that they misestimate the marginal values of the population time series, even more so the more the population fluctuates. This effect causes the LPI to spuriously decline by about 9.6%[15]. Buschke et al.[15] have also used a simple simulation model to argue that there is a fundamental asymmetry in the calculation of the LPI (see also ref. 16), as populations that fluctuate randomly and symmetrically from the same initial point reveal the decreasing LPI. Potentially, there may be multiple issues in the way the LPI is calculated, as well as in the data which are used for this calculation, that may lead to various biases and misunderstandings. It is thus worth exploring the LPI calculation in more depth.

Here we provide a detailed inspection of the methodological pipeline and computer codes used for calculating the LPI. We identify the sensitivity of the LPI to particular subjective decisions as well as potential methodological flaws in the calculation, some of them previously reported in the literature[10,11,14,15,17], but most of them unnoticed before. A thorough analysis suggests that accounting for these issues has the potential to weaken or even revert the trends of the LPI, altering the conclusions given by the Living Planet Reports[7,18]. We also point out that the major issues related to the LPI are not only caused by

the calculation itself, but are deeply related to the quality and representativeness of the underlying data.

## Results

### The LPI properly reflects the stationarity of the system

A recently published criticism of the LPI by Buschke et al.[15] has been based on the finding that the index declined even if the population trends were stable on average. Buschke et al.[15] derived the index value for simulated randomly fluctuating populations, where population changes adhered to a Poisson distribution with equal probability of being either positive or negative on arithmetic scale. Such populations diffusely diverged from one initial point (see Fig. 1 in ref. 15) and the whole set of all populations revealed the declining LPI. We found that the problem with this simulation strategy is that it leads to unrealistic non-stationary population size distributions. Even though the mean community abundance remains stable in the initial part of the simulation (50 years in Buschke et al.[15]), the population sizes steadily diverge, and community equitability thus decreases with time, the community being characterized by increasing difference between abundant and rare species. Moreover, such a process has an absorption boundary at zero, so all populations would eventually go extinct after a finite number of steps. The declining LPI in this case thus appropriately reflects the non-stationarity of the system when many species become rare, and those with increasing populations do not increase that much. In contrast, the LPI is stable whenever the distribution of population abundances is stationary and/or when population fluctuations are symmetric on logarithmic scale (Fig. 3). The symmetry on logarithmic scale makes a good sense, as population increase and decrease are essentially multiplicative processes, while symmetric changes in the arithmetic scale imply that the multiplicative increase of populations is lower than the multiplicative decrease (see also refs. 19,20). The LPI thus properly reflects symmetric multiplicative population changes, the decreasing LPI indicating asymmetry in this multiplicative population growth towards higher multiplicative decreases than increases.

A related criticism of the LPI by Puurtinen et al.[16] also highlighted the issue that population increases and decreases of the same magnitude on arithmetic scale lead to a decline in the index. The problem arises, according to the authors, from the LPI reliance on geometric averaging. According to the criticism, geometric averaging is appropriate only when population changes are averaged over a time series of interdependent values within a single population, but the LPI calculation instead averages trends across multiple populations for a certain year. But using geometric averaging for statistically independent measurements is not inherently flawed, given the multiplicative nature of population changes mentioned above. For example, populations that experience a doubling in size may be compensated in a stable system by populations that undergo a halving in size, ensuring that the LPI remains unchanged on average while total abundance may or may not increase for that particular year (see Fig. 1a in ref. 16). The criticism of geometric averaging is usually based only on a one-step change from year to year. However, in stable systems, where populations do not go extinct and do not experience systematic changes over time, both the LPI and mean abundance may show fluctuations, but may ultimately maintain stationarity (Fig. 3). Therefore, the LPI methodology can still offer a valid representation of the mean populations' stability.

### The effect of the number of records in the time series

According to the Living Planet Report[21], the duration of time series included in the calculation does not influence the LPI. However, while the length of time series (in years) and the number of time series used in the calculation indeed do not systematically affect the LPI, the number of records in time series does have an effect (Methods 'The effect of the duration and the number of records in the time series',

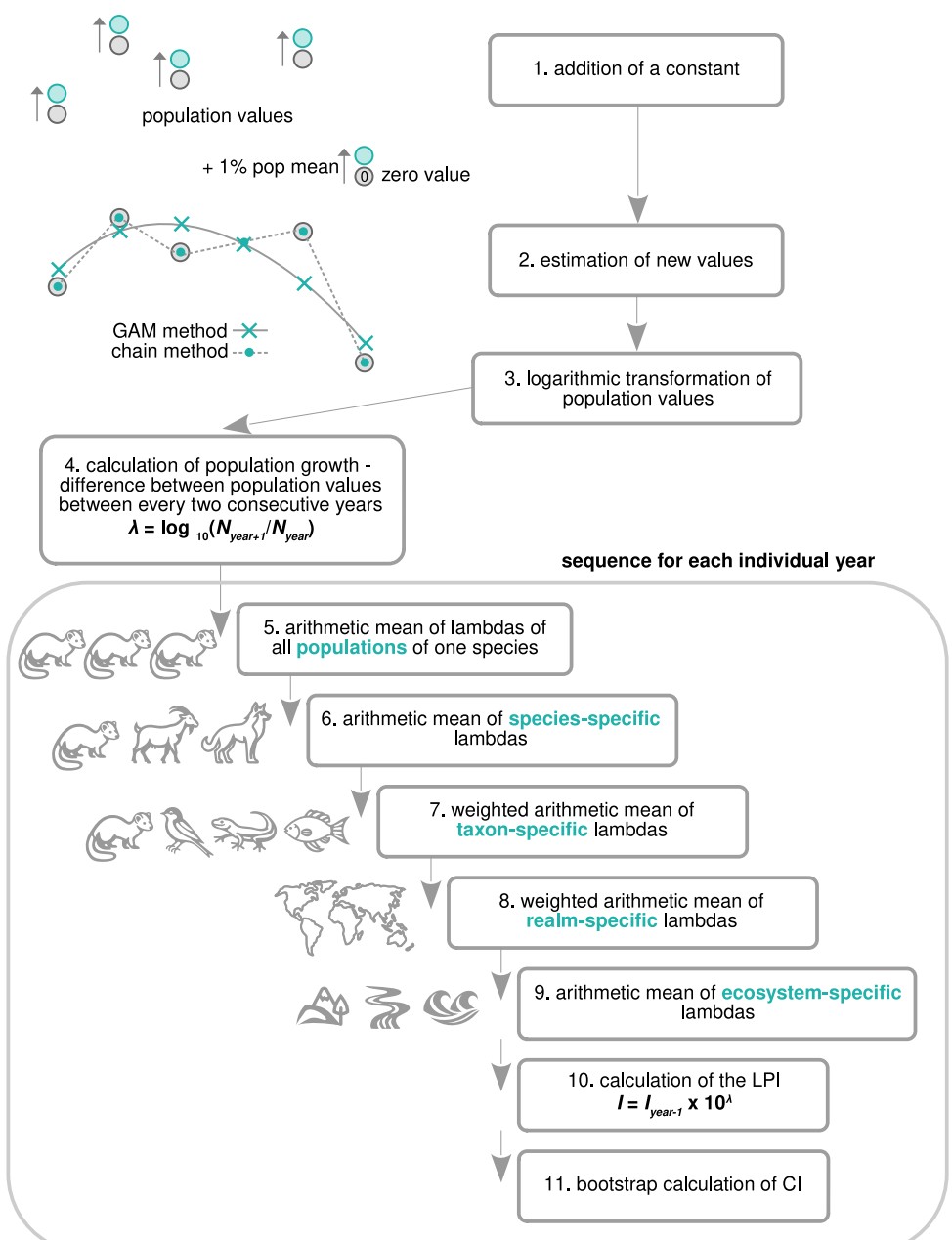

**Fig. 2 | Schematic description of the methodological procedure of the LPI calculation.** (1) Addition of a constant of 1% of the population mean to all values of the time series, if the time series contains zero in any year. (2) Estimation of the new population values by two methods; GAM method (time series >5 records) or chain method (time series <=5 records or if the GAM does not fit well). (3) Logarithmic transformation (using base 10) of the population values. (4) Calculation of the population growth rate ($\lambda$) as the difference between (logarithmized) population values between every two consecutive years = the logarithm of the ratio of population values $\lambda = log_{10}\left(N_{year+1}/N_{year}\right)$. (5 – 9) Sequence of hierarchical averaging of population growth across populations, species, taxa, realms and ecosystems for a single year. (10) Calculation of the LPI as $I = I_{year-1} \times 10^{\lambda}$. The index of the initial year 1970 is set to 1. (11) The bootstrap calculation of the confidence intervals of the index. For a detailed description of individual steps of the calculation, see Methods 'Calculating the Living Planet Index'.

Fig. 1, Supplementary Table 1, Supplementary Fig. 1, 2). The time series with fewer records tend to be declining on average (see also ref. [17]), which could be one of the reasons why some studies[8,9,22] that did not include these time series (less than 5, 6, or 10 recorded time points) did not show the prevalence of decreasing populations. Note that the existing LPI method also uses data of only two records per population, the exclusion of which reduces the decline in the index by 14.3% (Methods 'The effect of the duration and the number of records in the time series').

Including time series with the low number of records certainly has its merit, as they may be informative by themselves. However, one methodological factor may cause these time series to bias the LPI towards overall decrease. Population dynamics are affected by variability arising from demographic, environmental, and sampling stochasticity, which introduces uncertainty into the LPI[15,20]. To mitigate this uncertainty, a smoothing procedure based on the GAM method is applied during the index calculation. However, this smoothing model is applied only to time series longer than five records and only when it fits well (see Methods 'Calculating the Living Planet Index'). In contrast, short time series are processed by a method that does not employ data smoothing and thus retains the uncertainty, which has been shown to negatively bias the LPI[14]. Sampling errors act symmetrically on

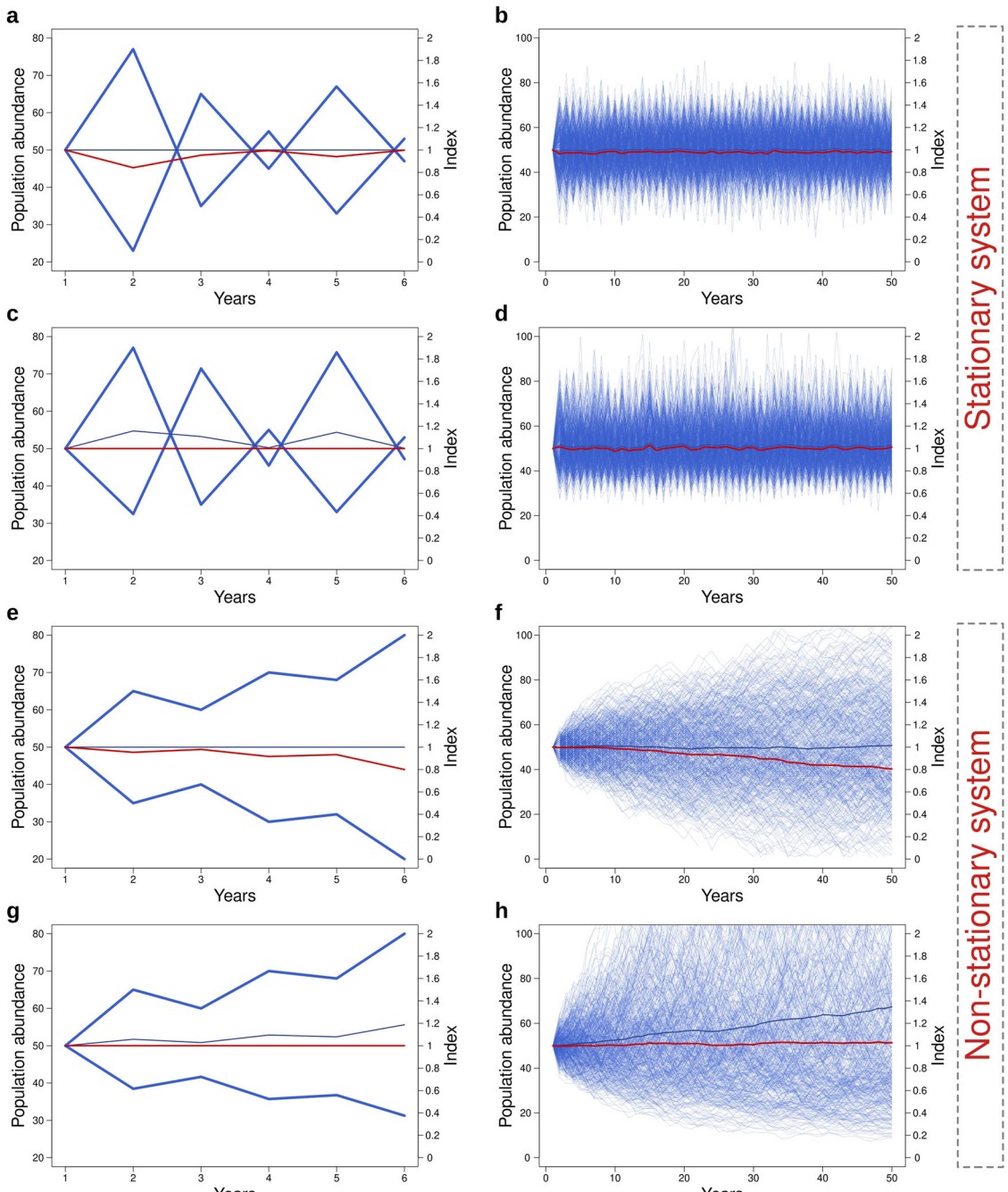

**Fig. 3 | Schematic description and simulations of the expected LPI under different scenarios of random population fluctuations.** Simulations (right column) of 500 populations (thin blue lines) all started with 50 individuals and were run for the period of 50 years. Random fluctuations can be either stationary (**a**–**d**) or non-stationary (**e**–**h**), and can be symmetric on either arithmetic (**a**, **b**, **e**, **f**) or logarithmic scale (**c**, **d**, **g**, **h**). In stationary systems (**a**–**d**), where populations fluctuate on arithmetic (**a**, **b**) or logarithmic scale (**c**, **d**), mean community abundance (central blue line), equitability, as well as the LPI (red line) remain stable. Note that the first scenario (**a**, **b**) can correspond to sampling errors of otherwise stable populations (it is modeled as gaussian deviations from a stable mean). Populations fluctuating randomly from the previous value based on Poisson distributions with lambda of 3 (**e**, **f**) (derived from Buschke et al.[15]) have a stable mean community abundance, but the decreasing LPI. The reason is that in such a case the population increases cannot compensate for population decreases on logarithmic scale. This system has an absorption boundary at zero, meaning that all populations would eventually go extinct. In contrast, a non-stationary system with symmetric fluctuations on logarithmic scale (**g**, **h**) reveals a changing mean community abundance, but the stable LPI, since the LPI expresses mean multiplicative population changes that correspond to symmetric fluctuations on logarithmic scale. The LPI is thus stable when populations do not go extinct and their sizes have a stationary distribution (**a**–**d**) or when their distribution is non-stationary, but population fluctuations have a multiplicative nature (**g**, **h**), corresponding to a random walk in the logarithmic scale. In contrast, the LPI systematically decreases in the case when population sizes diverge (the distribution is not stationary) and population fluctuations are symmetric in the arithmetic scale (**e**, **f**).

arithmetic scale and if populations are stable, the sample population sizes represent a stationary distribution and the index does not decrease. But short time series with sampling error typically increase or decrease overall, so they appear as involving non-stationary fluctuations on arithmetic scale. This leads, after averaging across multiple populations, to a decreasing index (Figs. 3, 4). This bias is particularly striking when populations values are low and discrete (Fig. 4).

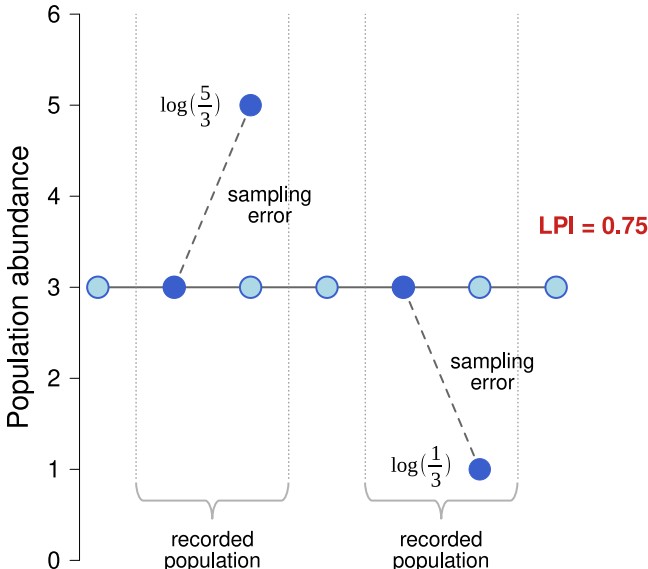

**Fig. 4 | Schematic description of a bias in the LPI of the population series with a few records.** Assume there is a stable population of three individuals (light blue circles). If such a population was surveyed in two separate time windows, it results in two population time series (dark blue circles) which may reveal increase and decrease, respectively, due to sampling error. Since sampling errors act symmetrically on arithmetic scale, the overall population growth is negative even if the increases and decreases are balanced—the LPI in the depicted case decreases from 1 to 0.75 (the change from 3 to 5 represents a multiplication by 5/3 = 1.66, while the arithmetically equivalent negative change from 3 to 1 represent a division by 3). The sampling-based increases and decreases of populations with a few records thus appear as non-stationary fluctuations even if the population is actually stable (see Fig. 3e, f). This bias is particularly pronounced when population values are low and discrete. The two-record populations represent a substantial portion of the data (see Supplementary Fig. 3) that may cause the LPI to decrease even if the sampling-based increases are balanced by sampling-based decreases. Note that it does not matter whether the two records occur in subsequent years or they are more distant in time.

Although the population series with the low number of records, and especially those with just two records, represent a non-negligible portion of the data and their removal would exacerbate geographical bias and reduce taxonomic and geographical representativeness (Supplementary Fig. 3, Supplementary Table 2), the problem of representativeness is accounted by weighting and thus is not as serious. In contrast, the bias produced by the sparse population time series in which the sampling error causes non-symmetric multiplicative population changes is substantial, and is responsible for a significant portion of the LPI decrease, as shown by removing these population time series (Supplementary Fig. 1, 2).

## Extreme sensitivity of the LPI to the initial decline of a few populations

The LPI is extremely sensitive to the availability of population time series at the beginning of the study period. It follows from the step-by-step calculation of the LPI, where the population change ($N_{year+1}/N_{year}$) is calculated between every two consecutive years and the index values are based on the multiplication of the previous value of the index by the geometric mean of population change (Fig. 2, Methods 'Calculating the Living Planet Index'). It means that the population increases/declines at the beginning of the time series transcribe through all the subsequent years. Since the population data from 1970s are sparse (Supplementary Fig. 4, 5), the low values of the LPI may easily result from few declining populations at the beginning of the study period. The LPI is presented in the arithmetic scale and in this respect it is asymmetric—since its value is calculated as the product of the previous year's value and the geometric mean of population change, the index

does not fluctuate much if the previous value is way below one even if the population growth rate is relatively high, while it may fluctuate considerably if the previous index value is high. An initial decrease of the LPI thus typically does not permit its later increase.

This effect is strengthened by the hierarchical averaging procedure—if some taxa are represented by only a few populations, these populations have the potential to disproportionately affect the global index. An extreme case comprises the herptiles in the Palearctic region, represented by only one (declining) population of viper *Vipera berus* for the period 1974-1977. Hierarchical averaging across taxa and biogeographical regions leads to the situation in which these four records of the viper population cause an 89.5% greater decrease (the index changes from the original value of 0.826 to 1.721 after removing these four records) in the final state of the LPI for the Palearctic realm (Fig. 5) and a 3.3% greater decrease in the LPI for the whole terrestrial system in comparison to the LPI without these four records (weighted in both cases). This viper population is the only single-population representative of the terrestrial system; for all the cases of single-population representatives of freshwater and marine ecosystems and their effects on the LPI, see Supplementary Notes. All the single-population representatives of population trends are situated at the beginning of the measurement period for a particular taxon and biogeographical realm. Although the end of the whole study period is also characterized by a lower number of population time series, this does not seem to have a substantial effect on the trend in the LPI, as restricting the population data to a particular end year does not change the final shape of the index. Although the latest version of the Living Planet Database (LPD) includes information which populations are no longer used for the global LPI calculation (including the abovementioned viper population), this does not solve the problem, as there are still cases in which a single population represents the entire taxon (e.g. again in Palearctic herptiles). Additionally, particular subsets of the data used for the calculation of the LPI for different taxa and/or regions will contain different single-population representatives. Moreover, the single representatives are the extreme case—if the beginning of the study period is represented by just two or three

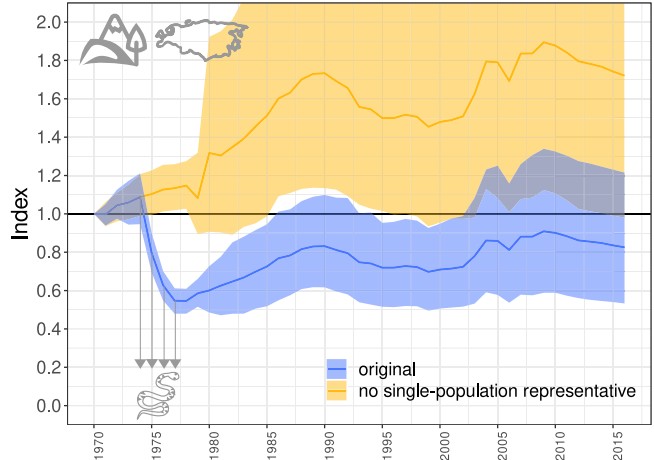

**Fig. 5 | The effect of a single-population representative of a whole taxon in a certain realm.** The original LPI for the whole Palearctic realm is blue and the LPI calculated without the 4 records (1974-1977) of one population of viper *Vipera berus* is yellow. The Palearctic LPI consists of the averaged population growth rate of three taxa (mammals, birds, herptiles), one of which (herptiles) was represented in 1974–1977 by only one declining population, causing the entire index to decline, subsequently affecting the remaining trajectory. Note that removing the effect of the four values of the viper, marked with arrows, changes not only the overall level of the LPI, but also its relative fluctuation, as low index values fluctuate less in the arithmetic scale by the very definition of the LPI. The colored area shows the confidence intervals and the lines show the LPI values.

populations, it can have a similarly disproportionate effect on the LPI as a single-population representative. This situation will persist with updated versions of the LPD, as the newly added population time series will most likely not overlap the beginning of the study period.

Note that the effect of the hierarchical averaging and under-representation of some taxa/realms depends on the grouping. If one (relatively smaller) group shows a significantly negative/positive population trend, this will strongly affect the resulting average of all groups. However, if this group is merged with another group, its negative/positive values are dissolved among all the values of both groups, and only then this merged group is averaged with other groups. For example, if we consider 5 realms and 3 taxa (in the unweighted form, as we compare it with the unweighted form of the next grouping) it decreases the decline in the terrestrial LPI by 6.3% (leading to the less decreasing LPI) compared to the situation when 6 realms and 4 taxa are distinguished (again in the unweighted form, as the weights are not available for this grouping) (Methods 'The number of biogeographical realms and vertebrate taxa').

The problem with the low number of population time series at the beginning of the study period cannot be easily solved. If we want to have the index for the whole period, we need to include all available time series, even if they are not particularly representative. In such a case, however, it is desirable to test the index for the sensitivity of their inclusion or removal, as well as for different ways of the grouping of taxa and realms.

## The problem of zeros in population time series

A conceptually more important issue is the way zeros are treated when calculating the index. The LPI is based on averaging the interannual growth rate $\log_{10}(N_{year+1}/N_{year})$, which cannot be calculated if the population size is zero in one of the compared years. This problem is solved in the LPI calculation by the replacement of zeros by a small value. In particular, a constant of 1% of the population mean is added to all values in the time series if any year contains zero (Fig. 2, Methods 'Calculating the Living Planet Index'). This is in fact, equivalent to a drop (in the case zero is at the end of time series) or an increase (if it is at the beginning) of the population size by two orders of magnitude, which is considerably larger than typical population fluctuations. Such population change is entirely arbitrary, and using a different proportion than 1% of the population mean would lead to very different interannual growth rate of a given population, and consequently a different LPI. If zeros were randomly distributed across population time series, this effect would cause just an increasing error, and not necessarily a bias towards the declining LPI. However, it is reasonable to assume that zeros occur with a higher frequency at the end of the time series, since populations are rarely studied when there are no individuals at the beginning. Such an asymmetry could cause the bias towards apparently declining populations. Indeed, the time series with zeros at the end outnumber those that begin with zero values in the Living Planet Database (Supplementary Table 3). Although the middle zeros or the middle sequences of zeros predominate overall, they cannot cause any bias.

To explore the extent of this effect, we recalculated the LPI with the removed zeros from all population time series (if zeros were in the middle of the time series, the series were split into multiple independent series; note that a significant number of population time series included sequences of several zeros; Supplementary Table 3). The change was substantial (Fig. 6, Fig. 1, Supplementary Table 1)—the decline of the global LPI was reduced by 19.2%, from the original drop to 32.7% (assuming 100% in 1970) to 51.9%—but differed among ecosystems. The reduction of the LPI decline was 33.8% in the case of the terrestrial ecosystem (from the original decrease to 36.8% to the decrease to only 70.6%), 19.3% for the freshwater ecosystem (from 18% to 37.3%), and less than 1% for the marine ecosystem (from 52.6% to 53.2%) (Supplementary Fig. 6, Fig. 1, Supplementary Table 1).

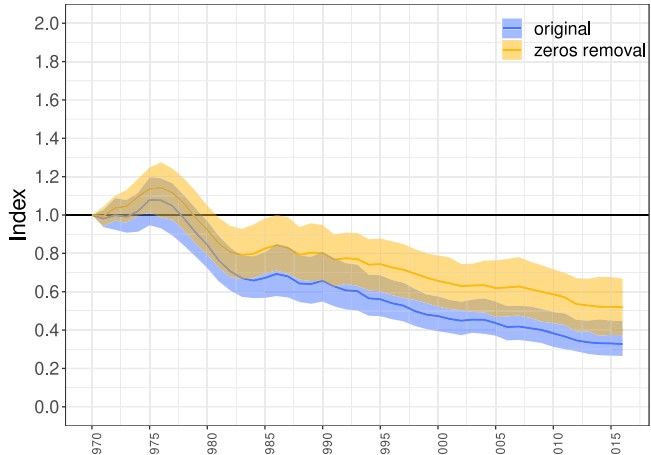

**Fig. 6 | The effect of removing zeros from population time series.** The original global LPI is blue and the LPI calculated without zeros in the population time series is yellow. The colored area shows the confidence intervals and the lines show the LPI values. For the effect of removing zeros for individual ecosystems, see Supplementary Fig. 6.

The differences between ecosystems appear to be due to the different prevalence of zero-valued ends of the time series (Supplementary Table 3). Importantly, removing zeros sometimes led to considerable broadening of confidence intervals, so that they overlapped the value of one, implying that it is often impossible to say with certainty whether there was any significant population decrease (Fig. 6, Supplementary Fig. 6).

Although the removal of zeros from population time series may look contentious, we argue that it is more appropriate than leaving the zeros in. Population fluctuations represent a process which is well characterized by the ratio of population sizes in consecutive time steps, corresponding to per-capita population vital rates that link population sizes in consecutive years. In contrast, colonization and extinction represent different processes which break this inter-annual link and thus cannot be mixed with population fluctuations even if the fluctuations sometimes do result in extinction. If a population is non-existent in one of the two years, population growth does not have any meaning. Replacing zeros with any value then arbitrarily modifies the link (or its absence) between population states in consecutive years and seriously distorts statistical properties of population fluctuations. Importantly, there is no justification for the claim that zeros should not be removed on the ground that this would change species composition across the time series (see refs. 23,24). As long as each population series is not bounded by a sequence of zeros throughout the whole study period, species composition changes regardless of whether zeros are excluded or included.

The zeros in the LPD certainly have various information values, some representing real extinctions, some emigration, and others just reflect sampling errors due to low population density or detectability. However, even if we could distinguish these situations, the methodology of the LPI calculation is unable to properly treat them. When the data undergo any transformation that involves division or logarithmization, the zeros can be only (1) replaced by a non-zero value or (2) removed. Even if the zeros are genuine and have a real basis, their replacement by any other value introduces a bias and statistical distortion due to the LPI algorithm (see refs. 23,24). The exclusion of zero values is thus essential for maintaining the integrity and accuracy of the population trends analysis. We do not dispute that ecologically meaningful zeros provide important insights into population dynamics, but they should be evaluated by an alternative analytical method alongside the LPI method (see refs. 8,24), e.g. by treating the colonization and extinction dynamics separately from population

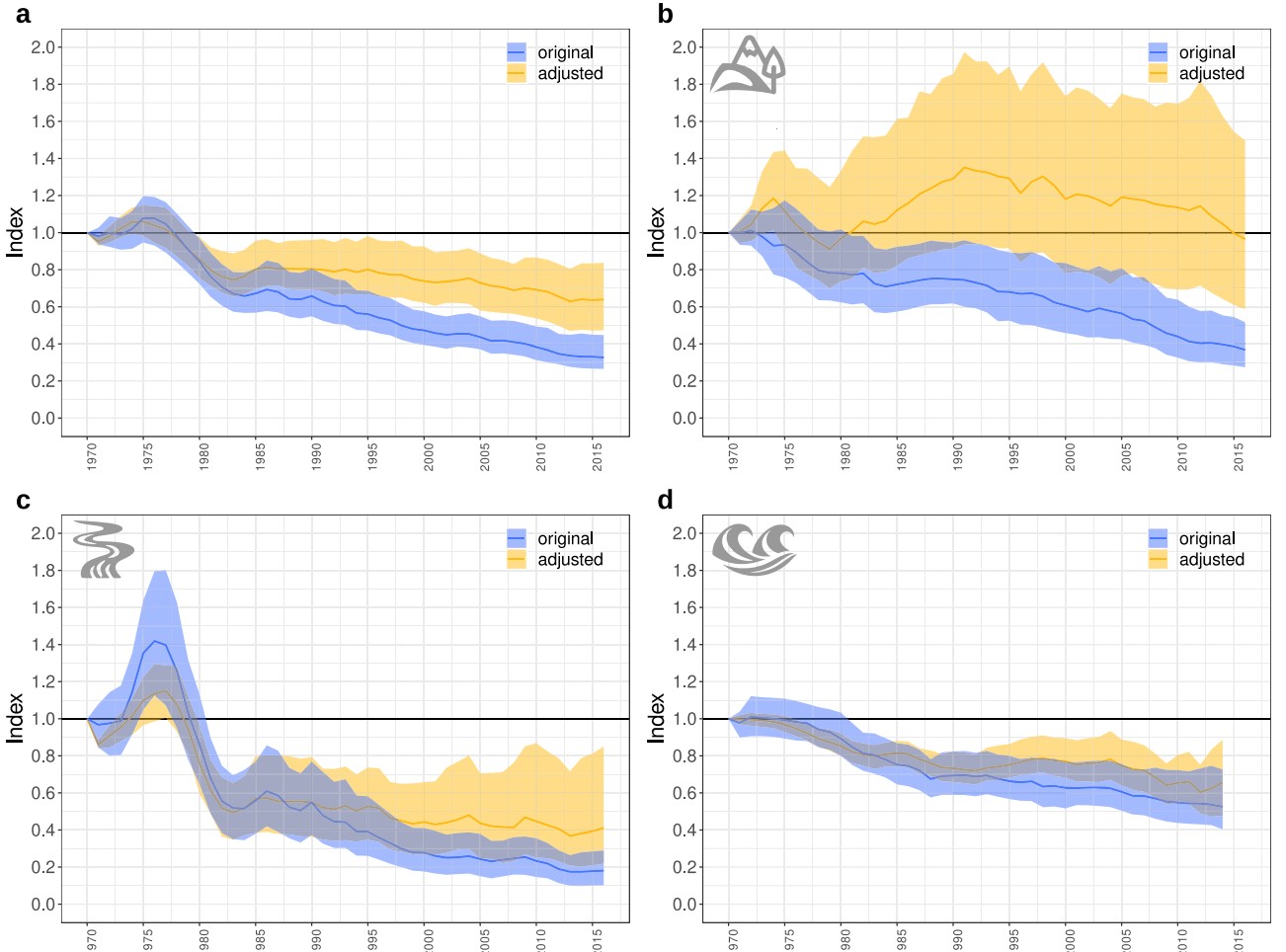

**Fig. 7 | The effect of adjustments of the LPI.** The original LPI is blue and the adjusted LPI is yellow, calculated without zeros in the time series of populations with at least 5 records globally (**a**) and separately for the terrestrial (**b**), freshwater (**c**) and marine (**d**) ecosystem. The colored area shows the confidence intervals and the lines show the LPI values.

fluctuations. Note that this problem of the presence of zeros in population time series goes far beyond the methodology of the LPI and concerns all studies that focus on population dynamics and need to deal with both population fluctuations (the realm of classical population biology) and colonization-extinction dynamics (the realm of metapopulation biology).

After we have accounted for the issues with the low number of records in some population time series and with the presence of zeros, we found that the adjusted Living Planet Index indicates considerably lower population declines than the original index (Fig. 7, Fig. 1, Supplementary Table 1; note the positive effects, depicted by green color in the tables, of the adjustments of the LPI).

## Discussion

We have shown that the LPI exhibits substantial variability depending on calculation settings. While these settings must necessarily at least partially depend on subjective choices, some of these decisions may compromise the robustness of the index. This problem is aggravated by the specificities of the data, namely the presence of population time series that have very few records, single-population representatives for some periods, taxa and regions, and presence of zeros. Due to the sensitivity of the LPI to subjective decisions and to specific problems with the LPI calculation, the LPI does not seem to accurately represent biodiversity trends. An indicator of the global state of nature should not be sensitive to the fact that 50 years ago one population of viper did not thrive well, and should not be affected by the particular way

population sizes were measured and how population absence was treated in the end or the beginning of the time series (see ref. 25). Similarly, a universal index of population change should not be sensitive to particular grouping to taxa and biogeographical realms if its aim is to provide a rigorous, repeatable indicator with a straightforward interpretation. These issues deserve particular attention if the LPI is calculated for individual regions or countries (see ref. 26), in which the effect of these biases may be even stronger than in the case of the global data.

Some of the issues mentioned above can be addressed in a relatively straightforward way. If an index that integrates population changes across regions and taxa into one number is desirable, it is necessary to find ways to make the index more robust and reliable. Since the calculation of the index inherently involves subjective settings, it is essential to thoroughly evaluate its sensitivity to these subjective decisions. The index calculation should thus encompass the full spectrum of reliable alternative decisions concerning grouping and weighting taxa and regions. Similarly, due to the fact that the geometric mean is strongly influenced by outliers, especially if the number of values entering the calculation is low, the index should use all variants of the removal of the single-population-representatives, i.e. sort of sensitivity analysis; see ref. 26. Other possibilities include shifting the reference year to limit the small number of populations at the beginning of the study period, reshuffling population time series within the study period (see also refs. 11–13,25,27). or weighting the data by the number of population records. It is also worth considering

whether it is appropriate to use time series shorter than 1-10 years or with less than 3-10 recorded time points (see refs. [12,13]) as they may not be able to adequately capture population trends[11,28], are subject to large sampling stochasticity and introduce uncertainty causing a downward bias[14]. Although removing sparse time series reduces taxonomic and geographical representativeness in terms of the number of utilized populations, it is less problematic than leaving them in the calculation, as they introduce a significant downward bias. Due to the way the index is calculated (i.e. by addition of the value of population change to the index value of the previous year), all process and measurement biases are accumulated at each step of the index calculation, increasing the inaccuracy of the LPI[14]. Additionally, the procedure of the LPI calculation based on averaging the interannual population growth rates is not compatible with the presence of zeros in the population time series, even if all zeros were ecologically meaningful. The more appropriate solution is thus not to include zeros. We are aware that the presence of zeros can be understood as an indication of population colonization or extinction, but these are essentially different processes from population fluctuations and should be thus treated separately alongside the LPI calculation (see refs. [8,24] for an example of how to do it, although these approaches also have their own limits and would also require a proper sensitivity analysis).

The LPI corrected for the biases explained above does not indicate as strong global population declines as the original LPI, published in the Living Planet Reports[7,18]. However, this does not necessarily mean that the situation is in reality better. Population time series in the Living Planet Database do not represent the results of a standardized systematic survey, but simply comprise all populations sampled for very different reasons (see refs. [29,30]). It is possible that the data do not include many populations that actually rapidly declined without even being documented, and ultimately disappeared—many habitats which were entirely converted to intensive agriculture, plantations or human settlements were not explored before the transformation, and are typically not studied after the transformation to document population disappearance. Many populations have been studied in pristine and/or protected areas, so the overall sample may be biased towards stable or increasing populations[31,32]. On the other hand, there may be some bias also in the opposite direction, stemming from the fact that ecologists typically begin to study populations which are already established and not those recently emerging[27], or directly focus on species at higher risk of extinction[33]. As the relative weight of these biases is hard to compare, the LPI corrected for all the calculation biases may still underestimate or overestimate real population changes.

The potential importance of these biases can be shown by the effect of weighting by taxon- and region-specific species richness. Although weighting was justified in order to reduce bias due to heavily oversampled regions, it introduces another bias as the weighted index is driven mainly by tropical populations (species-rich regions)[11]. These regions might be sampled for specific reasons and at specific locations. The declared advantage of the Living Planet Database, namely the high number of time series collected, cannot overcome the fact that the database is a heterogeneous set of spatially and temporally biased data collected for different purposes. A solution to this problem would be to use only population time series from standardized systematic surveys (see ref. [25]) where all populations have been sampled regardless of their size, trends, and environmental changes, but such studies are rare and (again) strongly geographically biased[34–36].

Even at the regional or national level where more balanced sampling may be ensured, the data should not be sensitive to weaknesses in the LPI methodology (including errors in the code; see Methods 'Calculating the Living Planet Index' and Supplementary Notes), otherwise this method may produce even greater inaccuracies in population trend estimates than in the case of a larger dataset, as we

show for specific taxa and regions. The LPI methodology and datasets are already used to evaluate biodiversity changes and to monitor compliance with various international agreements and targets tailored to specific regions and groups (e.g. the LPI for Belgium, Netherlands, China, Austria, Mediterranean wetlands, migratory species, freshwater megafauna, reptiles, specific biomes; see ref. [26]), and it is questionable to what extent the resulting indices are distorted by the abovementioned shortcomings.

In summary, we have shown that there are issues with the calculation of the Living Planet Index that lead to an overestimation of vertebrate population declines. Some of the biases can be corrected in a straightforward way—for instance, there have been small errors in the code which can cause serious bias in smaller datasets, and we provide the corrected code (Supplementary Software). The zeros need to be removed from the population time series and the LPI method needs to be accompanied by an analysis of extinction/colonization (see ref. [8]) or a method capable of processing zero values should be used (see ref. [24]). It is also necessary to account for the uncertainty introduced by the data by including time series of a reasonable number of population records to avoid downward bias. Additionally, a rigorous approach should include the exploration of the consequences of changing biogeographical/taxonomic groupings and the exclusion of sparsely populated taxa, to assess the sensitivity of these alternatives on the resultant index. Even after accounting for all abovementioned issues, the reliability of the LPI may be compromised by the heterogeneity and unequal geographical cover of the data (see ref. [11]), suggesting the need for standardized survey data collected independently of specific purposes or locations.

## Methods
### Calculating the Living Planet Index
We have explored the code used for the calculation of the LPI. Although Loh et al.[1], Collen et al.[2] and McRae et al.[3] provide the basic principle of calculating the LPI, the exact methodological procedure is clear only from the code of the package rlpi (v.0.1.0) in R[37]. This package was created and made available by the Zoological Society of London in 2017 and presented in McRae et al.[3], who also introduced the diversity-weighted form of the LPI. In fact, without the precise procedure it is not possible to replicate the calculation to obtain the LPI identical to the one presented in the Living Planet Reports[7,18]. We are aware of the possibility that an updated version of the code exists, but we have been confined only to the material that is publicly presented and recommended for use.

The procedure consists of several steps; addition of a constant to the whole time series if it contains zeros, estimation of new population values by the GAM or chain method, calculating mean population growth of each population for each year, and hierarchical averaging of population growth from populations to species, taxa, biogeographical realms and ecosystems (Fig. 2; see below for the detailed description and Methods 'The Living Planet Database' for a description of the database used). In a detailed R-code inspection we found errors in the original calculation of the LPI; see Supplementary Notes for their complete list and R-scripts with marked errors (Supplementary Software). All calculation errors in the code have a negligible effect on the final shape of the global LPI trajectory, but are evident in some cases where the LPI is calculated for a smaller subset of data—a certain taxon or biogeographical realm (Supplementary Fig. 7). We provide the R-code with all errors corrected (Supplementary Software).

The methodological procedure for calculating the LPI consists of these steps:

1. Addition of a constant of 1% of the population mean (the mean from all non-zero values) to all values of the time series if the time series contains zero in any year. If a population series contains only zeros, the added constant is $10^{-17}$ (we removed these cases).

2. Estimation of the new population values by two methods (also the way to estimate missing values, i.e. values for years without population records):

GAM method is used if the length of the time series is equal to or longer than 6 records and only if the GAM fits well. The GAM smoothing parameter is set to 1/2 of the length of the time series. The GAM method is implemented on logarithmic (base e) values and the values estimated by the model are subsequently delogarithmized.

Chain method is used if the length of the time series is less than 6 records or if the GAM does not fit well (or if all population values are the same). It is a log-linear interpolation for missing values in the population series (see Equation 2 in Collen et al.[2]).

3. Logarithmic transformation (base 10) of the population values.

4. Calculating the difference between the (logarithmized) population values between every two consecutive years = the logarithm of the ratio of population values = population growth = lambda:

$$\lambda = log_{10}\left(N_{year+1}/N_{year}\right).$$

5. Calculating the arithmetic mean of lambdas (the logarithm of the geometric mean) of all populations of one species within one biogeographical realm (for an individual year). There are 5 (for the terrestrial and freshwater ecosystem) or 6 (for the marine ecosystem) biogeographical realms distinguished (see below).

6. Calculating the arithmetic mean of species-specific lambdas across all species of one taxon within one realm (for an individual year). There are 3 (for the terrestrial ecosystem) or 4 (for the freshwater and marine ecosystem) taxa distinguished (see below).

7. Calculating the weighted arithmetic mean of taxon-specific lambdas across all taxa within one realm (for an individual year). The taxon-specific lambdas are weighted by the ratio of the species richness of a given taxon and the species richness of all the taxa together (the weighted method was implemented by McRae et al.[3]).

8. Calculating the weighted arithmetic mean of realm-specific lambdas across all realms (for an individual year). The realm-specific lambdas are weighted by the ratio of the species richness of a given realm and the species richness of all the realms together (the weighted method was implemented by McRae et al.[3]). The result is one lambda for a certain year.

9. Calculating the arithmetic mean of ecosystem-specific lambdas across all ecosystems (for an individual year) is obtained by dividing the realm-specific weights by the number of ecosystems (only in the case when the global LPI is calculated), i.e. all the realm-specific weights are multiplied by 1/3 (this procedure is not implemented in the code).

10. The calculation of the LPI as $I = I_p \times 10^\lambda$, where Ip is the index of the previous year and the index of the starting year 1970 was set to 1.

11. The bootstrap calculation of the confidence intervals of the index. The method involves 100 resamplings of species from each taxon with replacement.

The last 7 steps run in a loop for each year.

More formally, the global LPI is calculated as a hierarchical sequence of five geometric means:

$s_x = (\prod_{i=1}^{n} p_i)^{1/n}$, i.e. a geometric mean of $n$ ratios ($p$) of population values ($N$) of two consecutive years ($p_i = N_{year+1}/N_{year}$) of the $x$th species,

$t_q = (\prod_{x=1}^{d} s_x)^{1/d}$, i.e. a geometric mean of $d$ species-specific $s_x$ of the $q$th taxon,

$r_b = (\prod_{q=1}^{c} t_q)^{1/c}$, i.e. a geometric mean of $c$ taxon-specific $t_q$ of the $b$th realm,

$e_y = (\prod_{b=1}^{f} r_b)^{1/f}$, i.e. a geometric mean of $f$ realm-specific $r_b$,
$g = (\prod_{y=1}^{h} e_y)^{1/h}$, i.e. a geometric mean of $h$ ecosystem-specific $e_y$.
The global LPI is then calculated as $I = I_p \times g$,

therefore it is the product of the previous year's index ($I_p$) and $g$. It holds that the logarithm of $g$ is the final lambda ($\lambda$)−the mean population growth rate of all populations. The index can be thus expressed as $I = I_p \times 10^\lambda$.

In practice, the calculation is performed as a hierarchical sequence of arithmetic means, where in the first step the logarithm of the ratio of population values is averaged arithmetically. The arithmetic mean of logarithms is equivalent to the logarithm of the geometric mean, thus $\lambda = log_{10}(g)$.

The R-function from the package rlpi (https://github.com/Zoological-Society-of-London/rlpi) allows various calculation settings of the LPI. It is possible to change the minimum length of the time series (the number of records, but not the number of years) included in the calculation, the constant replacing zeros, the length of the time series for which the GAM or chain method is used, the GAM smoothing parameter, the limit value for outlying lambda and whether to replace the outlying lambdas, and the use of weighting. The weights of particular taxa and realms were obtained from McRae et al.[3].

The shape of the LPI curve is mostly influenced by two parameters; the number of records in the time series (fullness) and the use of weights (see ref. 3). The difference between the weighted and unweighted form of the global LPI is 44.5% (much greater decline in the weighted than unweighted form). The effect of weighting for the terrestrial, freshwater and marine LPI causes a 14.8%, 47.3% and 83.5% greater decline, respectively, in the weighted than unweighted form (Supplementary Table 1).

## The effect of the duration and the number of records in the time series

The original method of calculating the index takes into consideration all time series longer than one record (2 or more). If the global LPI is calculated only with the time series with at least 3, 5, 10 records, the decline in the index is reduced by 14.3%, 14.7%, and 26.4%, respectively (Supplementary Table 1, Supplementary Fig. 1). In the case of the terrestrial LPI, the inclusion of only time series with at least 5 records causes a 5.5% reduction in the decline. If the freshwater LPI is calculated with time series equal to or longer than 5 records, the decline in the index is reduced by 14.2%. Similarly, for the marine LPI, the decline in the index is reduced by 25.6% (Supplementary Table 1 for all 3/5/10-records options, Supplementary Fig. 2). However, the length of the time series of estimated values may not correspond to the number of population records in the time series. If the individual records are not consecutive in each year, the missing values are calculated (by the GAM or chain method). Therefore, it can happen that a time series having five records can enter the index calculation as a time series of more than five estimated population values, i.e. longer than four years. In any case, the number of the records in the time series (adjustable parameter in the R-code) limits the minimum length of the time series, i.e. its duration in years (which is not an adjustable parameter in the original code). On the other hand, the length of the time series (the duration) does not affect the minimum number of records, as it can be always as few as two records. A relatively smaller decline in the index after removing time series with fewer records would suggest that time series with lower fullness (as defined here) are on average those comprising decreasing populations, but this may be an effect of sampling errors (Fig. 4). In contrast, the length of the time series (the interval between the first and last observation) has very little effect on the overall trend (Supplementary Table 1, Supplementary Fig. 1 and 2; see also ref. 38). We tested a simple removal of population time series based on a selected number of records/years, because such an extremely heterogeneous dataset of taxa/species and abundance proxies

does not allow the establishment of unequivocal criteria for the minimum number of records needed to detect a population trend.

The index calculation includes a smaller number of populations when limited by the duration of the time series (19,205/16,555/12,660 populations considered for at least 3/5/10-year-long time series). Even fewer populations are included when the limitation is based on the number of records in the time series (17,753/13,868/9,528 populations considered for at least 3/5/10-record-long time series) (see also Supplementary Fig. 3, Supplementary Table 2). However, the resulting index is affected only by the limit on the number of records in the time series. This suggests that the LPI does not demonstrate a systematic trend based on the number of population series utilized and duration of time series, but it does reveal a trend influenced by the number of records within the time series.

### The Living Planet Database

The data for the LPI calculation was obtained from the Living Planet Database (LPD) (https://livingplanetindex.org), which currently includes freely available time series data since 1970 to the present (the data on many realms and taxa are there only until 2014) for 22,175 populations of 4,777 mammal, bird, reptile, amphibian and fish species from terrestrial, freshwater and marine ecosystems (data downloaded at 5/2021 and 1/2022; all data were updated to 1/2022) (Supplementary Table 4). The LPD is repeatedly updated with new population time series throughout the considered time frame; as a result, each new round of the LPI calculation works with a different data collection. The basic data units (records) are population sizes or various proxies of abundances (e.g. the number of individuals, breeding pairs, eggs, the number of burrows) or population densities or biomass (based on pitfall or camera traps, weight of net catch, various records per area or time) for different years. The population time series begin and end in different years and the records were sampled with different frequencies and often irregularly. The original LPI calculation considers 5 biogeographical realms and 3 taxa for the terrestrial ecosystem, 5 realms and 4 taxa for the freshwater ecosystem, and 6 realms and 4 taxa for the marine ecosystem (see SI in McRae et al.[3]).

### The number of biogeographical realms and vertebrate taxa

In the LPD, there are 6 biogeographical realms distinguished for the terrestrial/freshwater ecosystem; Afrotropical, Palearctic, Nearctic, Neotropical, Australasia and Indo-Malayan. The alternative is that Australasia and Indo-Malayan realms are merged into the Indo-Pacific. For vertebrate taxa, 5 groups are distinguished; birds, mammals, fish, reptiles and amphibians. Reptiles and amphibians can be merged into one group of herptiles. For the marine ecosystem there are 6 realms; Arctic, Atlantic North Temperate, Atlantic Tropical and Subtropical, Pacific North Temperate, Tropical and Subtropical Indo-Pacific, South Temperate and Antarctic. As there were weights for only 5 terrestrial/freshwater realms (Australasia and Indo-Malayan as one Indo-Pacific realm) and 3 and 4 taxa, respectively (reptiles and amphibians as herptiles), it was necessary to use the merged alternatives.

### Reporting summary

Further information on research design is available in the Nature Portfolio Reporting Summary linked to this article.

## Data availability

Data of population time series stored within the Living Planet Database are managed and maintained by the Indicators & Assessments Unit at the Zoological Society of London (ZSL) and WWF International (WWF) and available on their website (https://livingplanetindex.org/data_portal). The downloaded data (1/2022) included the entire publicly available collection of population time series of vertebrate species from around the world. The terms of use for data from the Living Planet Database are set out in the Data Use Policy (https://livingplanetindex.org/documents/data_agreement.pdf).

The values for weighting individual groups are available in Supplementary Tables S10-S13 from McRae et al. (2017).

## Code availability

The open-source code used to calculate the Living Planet Index from LPD data is contained in the R-package 'rlpi' (v 0.1.0), developed and maintained by the Zoological Society of London (ZSL) and available from the GitHub repository: https://github.com/Zoological-Society-of-London/rlpi. R-code and outputs (R-scripts and RData files) for all analyses used for this study are available in Supplementary Software.

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

## Acknowledgements

The research was funded by the Czech Science Foundation (GAČR 20-29554X).

## Author contributions

A.T. and D.S. conceived the study and wrote the manuscript. J.S. helped with writing, editing, and discussing ideas. A.T. compiled the data, processed the computer code, and performed the analyses.

## Competing interests

The authors declare no competing interests.
