## [Peer Review File · Nature Communications]

Mathematical biases in the calculation of the Living Planet Index lead to overestimation of vertebrate population declineReviewer #1 (Remarks to the Author):

This submission presents a sensitivity analysis of the Living Planet Index (LPI) to make four main statements: (1) the LPI represents the stationarity of the system accurately; (2) the LPI is affected by population time-series with few records; (3) the LPI is sensitive to the initial decline of a few populations; and (4) the way zeros are handled affects the LPI. The authors use these four results to support a conclusion that “population time series used in the Living Planet Database cannot provide a reliable picture of current biodiversity changes” (lines 24-25). Overall, while this submission presents several technically correct results, I am of the view that it misinterprets these results to draw unsubstantiated conclusions.

My main issue is philosophical. It is not stated as plainly as this, but this submission implicitly presumes the existence of a normative indicator of global population trends, an empirical gold standard representing the ‘true’ state of populations. Any deviation from this unstated ideal is subsequently interpreted as a methodological flaw or a mathematical bias. To me, this is an informal logical fallacy known as the Nirvana fallacy (https://en.wikipedia.org/wiki/Nirvana_fallacy).

The LPI is an index – a quantitative summary of a much more complex natural phenomenon – in much the same way as other imperfect, yet still useful indices. The S&P500 is a composite index of US stock market. GDP is a composite index of a country’s economic production. The Human Development Index is a composite index of a country’s development status. We accept that each of these indices are useful without being perfect. Companies go bankrupt even when the S&P500 index rises, but we do not interpret this as evidence that the index itself is mathematically flawed or biased.

I believe the authors make a fundamental mistake when they dismiss the analytical process of the LPI as ‘arbitrary’ (line 250). Some steps in the analysis may be ‘subjective’, but this is not the same as being arbitrary. The choices around including short time-series, weighting taxa and regions, averaging across realms and systems, and handling of zeros were not taken at random, but were judgements based on analytical trade-offs. Making different analytical decisions does not universally improve the LPI, it simply introduces a different set of trade-offs (which I describe below).

The authors end their manuscript with the sentence “Relying on the LPI, even after the modifications we propose, provides incomplete information about the state of global biodiversity” (lines 335-337). This is a trivially true statement, and one that applies to any indicator of global biodiversity. Similarly, they write (lines 76-77), “We also point out that the major issues related to the LPI are not only caused by the calculation itself, but are deeply related to the quality and representativeness of the underlying data”. Again, this is trivially true for any dataset that is not a representative random sample of the population it is drawn from (i.e., almost every dataset of global diversity).

In the first sentence of this review report, I purposely refer to this study as a ‘sensitivity analysis’ of the LPI because, to me, that is all that it is. It explores how the LPI responds to incremental adjustments to its calculation. This is a noble effort, for which the authors should be commended, but it is distinct from proving that the LPI is flawed or invalid.

What follows are specific comments on the four main findings of this study.

1. The LPI properly reflects the stationarity of the system

This section is technically correct. If a population is stationary and fluctuates randomly around that stable equilibrium, then the LPI is unbiased. By contrast, when populations fluctuate on a random walk, then the LPI is downward biased because individual populations diverge from the average response. This raises the obvious question of how populations are more likely to fluctuate, a question that has no readily available answer. I assume that sampling error would likely fluctuate around a stationary value (because an individual missed in a survey one year can still be recorded in the next); but population stochasticity would likely cumulate through time in a non-stationary

way (because random births/deaths affect population dynamics in subsequent years).

In this context, the authors may be interested in the comment (Talis & Lynch, 2023 <https://doi.org/10.1038/s41559-023-02086-w>) and response (Buschke et al. 2023 <https://doi.org/10.1038/s41559-023-02087-9>) published earlier this year. What matters to the LPI is not the relative frequency of different types of randomness (i.e. sampling error, environmental stochasticity, demographic stochasticity), but rather the magnitude of the form of randomness that does affect the index (i.e. demographic stochasticity)

2. Number of records in the time-series

The authors are plainly correct that the LPI will be different if one removes time-series with few records. What is missing, in my opinion, is an assessment of how removing these shorter time-series affects taxonomic and geographic representativeness. I worry that there is a risk that using only high-quality time-series would turn the LPI into an index of birds from USA and Europe, possible in protected areas (which are more likely to be monitored over long periods, but less likely to show declines).

I checked the 2022 public version of the Living Planet Database and saw that the average number of records per time-series Africa (7.4) and Latin America (6.7) is much less than Europe (14.6) and North America (18.6). Similarly, birds (14.9) tend to be surveyed for longer than amphibians (5.7) and even mammals (7.6).

I am not claiming that removing short time-series is necessarily wrong and that the LPI is right, but rather am illustrating the point that addressing one bias in the LPI (removing short time-series) can inadvertently introduce a different bias (taxonomic and geographical).

An alternative argument – and one that I personally support – is that these results should not justify the exclusion of short time-series in the calculation of the LPI, but should rather stimulate the collection of better quality data for those populations with inadequate sampling. My view is that unintuitive results should encourage action to improve data, not disregard what limited data we already have. (I acknowledge that my interpretation is subjective, but it is an equally valid interpretation of the authors' results).

3. Extreme sensitivity of the LPI to the initial decline of a few populations

This result has been presented before by other studies (by Brian Leung and those commenting on his paper), so I have no reason to doubt it. However, it presents another example of, what in my view, is only one possible solution to a real problem. If the LPI is sensitive to a few anomalous trends from poorly studied taxa from understudied parts of the globe, the onus should be on collecting better data, not removing what little data there is. To me, removing messy data from poorly studied regions and taxa makes the LPI less representative of global biodiversity, not more.

The caption to Figure 2 implies that there were three taxa, only one of which (*Vipera berus*) was declining. If this is the case, then the question can be posed to the authors why they are confident that the two remaining taxa are more representative of amphibians and reptiles in the Palearctic? (i.e., how can we be sure that these taxa are not resilient species that are more likely to be studied?)

But there may also be a technical point of confusion here, because I only have access to the public Living Planet Database from 2022 and it includes 54 time-series from 25 species of amphibians and reptiles in the Palearctic. In this version of the data, *Vipera berus* has data from six different populations (from three countries). The one population from Sweden, which declined between 1974 and 1981 is marked in the metadata as having been excluded from the final calculations (I am unsure why, it seems to be an island population, so highly variable population sizes seem realistic). Nevertheless, it seems that this section (lines 157-172) is based on an outdated version of the Living Planet Database and might no longer be supported by empirical evidence.

4. The problem with zeros in the population time-series.

The authors highlight an important issue about zeroes, but I wonder if they could have made more of an effort to understand the zeroes in the Living Planet Database. I took a quick look at the public version of the database from 2022 and found that of the populations with zero values, 66% were birds, 24% were ray-finned fish, and 8% were mammals. The survey methods for fish were mostly catch data, where one could assume that low catch rates are due to reduced population densities or at least changing size frequencies that are less likely to be caught in larger net sizes (in which case, removing zeros would downplay these population reductions/changes).

Equally interesting is that 78% of the bird populations containing zeros seem to come from the Australian Shorebird Monitoring programme. I am not familiar with this program, but it seems to be a citizen science program that records shored birds twice a year. It is unlikely that these zeros represent extirpations and recolonisations, but rather migratory species that either were missed in surveys (presumably due to lower densities) or did not appear during the survey period (presumably due to the vagaries of shorebird migration). In both instances, it could be argued that these zero values are ecologically informative. This raises an issue not considered by the authors about whether the LPI should only consider closed populations (i.e. those unaffected by migration dynamics).

In summary, I think this is another example where the authors are too quick to advocate for a specific analytical choice, instead of reflecting on whether exploring the imperfect LPI can yield deeper ecological insights about the state of biodiversity.

To conclude, I am of the view that the authors draw the wrong conclusions from their otherwise technically correct analyses. The shortcomings of the LPI and the Living Planet Database should not discourage their use, but should rather stimulate action to improve these resources.

Finally, I am certain the authors agree with me (and the originators of the LPI) that global vertebrate populations cannot be summarised in a single metric. Instead, the value of the LPI is to provoke action to collect better data or explore nuanced analyses. This is demonstrated through the consistent growth in the number of population time-series in the Living Planet Database over the last two decades. To pre-empt any potential concerns that summarising the LPI in a single metric can create the wrong public impression, I would like to point out that the 2022 version of the Living Planet Report was also covered in the media in a very different way. High profile pieces in the New York Times (<https://www.nytimes.com/2022/10/12/climate/living-planet-index-wildlife-declines.html>), Vox (<https://www.vox.com/down-to-earth/2022/10/12/23399105/biodiversity-loss-wwf-living-planet-index>), and Smithsonian Magazine (<https://www.smithsonianmag.com/smart-news/animal-populations-faced-a-very-sharp-decline-since-1970-180980957/>) made a concerted effort to explain the nuance of the LPI and how it might be interpreted.

I think this is a very valuable development, and it shows the usefulness of having a single metric as a gateway to deeper reflection and public education.

Reviewer #2 (Remarks to the Author):

This ms takes by far the deepest dive into the LPI to date and finds some pretty substantial problems.

First, LPI is probably the single most widely known and used index of human impact on biodiversity available. As noted in the first paragraph this has gone from an index that receives exceptionally wide press coverage to one that is embedded into UN conventions and treaties. Thus understanding its strengths and weaknesses is critical and entirely NatureComm appropriate.

At the same time my read of the conservation community is that they tend to not trust the LPI and see it as badly exaggerated without knowing why. Several recent papers have hinted at why there

might be problems. This paper reviews prior efforts and then takes a much deeper dive to arrive at strong answers. Several methodological issues are discussed (e.g. zeros) but the real problems appear to lie in the nature of the data (groupings and weightings with sparse coverage, zeros in the data, etc).

To be honest my conclusion after reading this paper is that the LPI is untenable as a policy-relevant index and cannot be fixed. Others might not draw quite as strong a conclusion, but this debate needs to be had.

The paper is clear and scientifically correct. I have relatively little suggestion for improvement.

My one suggestion is that the finding that length of time series (by # of records) is important with short time-series being biased to decreases, which has been noticed more glancingly by a few other papers (e.g. Leung 2019 and the commentaries). I did not see a clear explanation (either conceptual or data) why the short time series are biased down - it is not obvious that this has to be. Anything the authors can add to this debate would help make this a "one-stop" paper on the LPI critiques.

Reviewer #3 (Remarks to the Author):

This manuscript focusses on the LPI for biodiversity trends estimated for vertebrates from a large collection of (biologically and statistically heterogeneous) time series. The authors raise some concerns about data quality that are echoed elsewhere, but they also make some points about the computation methods and assumptions that go into estimating the index.

Overall, I think the manuscript makes some good points, but I feel some of these points are more general than to the LPI, and this be made more clearly than in the current version of the text. For example, potential biases caused by not including measurement error, immigration/emigration processes would affect any method of trend estimation, and not just the LPI. There is a danger that by focussing very narrowly on the LPI, some of these wider points are not absorbed by the more general reader. My suggestion would be to try and widen the scope of the points raised, to go beyond the LPI.

Early on it is pointed out that different methods lead to different conclusions regarding biodiversity trends (refs 8 and 9 are used here). However, it is fair to say there will be methodological biases in these approaches too because they use simplifying assumptions. For example, ref (9) does not appear to include process error in their trend estimates, nor is it clear any model validation is done. My point is that some of the issues raised in this manuscript are more general than to the LPI.

The manuscript results section begins with the rebuttal of some recent criticisms of the LPI analyses regarding the estimation of mean trends. I very much agree with the points raised by the authors of the current manuscript, but I do feel that this section should be supported by some analyses (figures and, where appropriate, maths and/or code). This would strengthen the argument in lines 86-87: "We found out that the problem with this simulation strategy is that it leads to unrealistic non-stationary population size distributions."

The effect of the number of records in the time series.

Whilst I agree with many of the points made here, I feel there are other considerations. A key question for me is also whether short time series are expected to be biased? I expect them to be noisy, but, if we simulate a large number of trends with the same underlying trend, do we expect a bias from the averaging of these short times series? The data-poor time series might be declining because there is a bias towards monitoring declining species, but this is a bias created by the non-random species/population sampling rather than a bias from short (data-poor) time series per se. It is good to point out how including/excluding the data poor time series changes the LPI, but it should open more of a discussion about what is going on in these populations, and how we should treat the data associated with them. It isn't clear that a one size fits all approach is the best (ie

including or excluding all data below a certain length).

I would also suggest the data poor time series could be weighted. Indeed this is what has been done in this manuscript -short time series are effectively given a weighing of zero in supplementary figure 1 etc. But there might be other useful ways to weight time series that include as much of the data as possible, but acknowledges uncertainty in the individual trends based on the number of sampled time points. This is similar to a meta-analysis approach, and would very likely place the estimated biodiversity trends in between the LPI and adjusted-LPI.

Like many of the issues raised in this manuscript, these are decisions all methods need to consider when estimating general trends/indices of biodiversity change.

Extreme sensitivity of the LPI to the initial decline of a few populations

This section raises similar issues to the last one, and I think the authors make a good point about taxa being represented by just a few populations -this point should also apply to the previous section.

The problem of zeros in population time series

This section discusses how biases may arise in the LPI calculation because the growth rates are log-transformed, and when there is a 0 this will cause well-known problems. As pointed out the LPI uses one of several options. I think the other options should be made explicit.

In my view this problem highlights a wider problem -the populations are generally not closed. Any time point that has a 0 and is followed by a non-zero number shows that either (i) there is observation/measurement error; or (ii) there is immigration/emigration in this system. I don't think the authors consider observation error in their arguments, but it could explain many 0's. However, if there is immigration/emigration then other time points will be biased too, assuming the model does not take these into account. Even without the $\log(0)$ problem, trends could be biased if they do not take these processes into account, and so the problem goes beyond considering only the 0 time points ie dealing with zeros in a different way does not get round the point raised about immigration/emigration processes.

Finally, I agree that the coding errors reported are important to have catalogued so they can hopefully be addressed if the LPI is used further down the line.

REVIEWER COMMENTS

Reviewer #1 (Remarks to the Author):

This submission presents a sensitivity analysis of the Living Planet Index (LPI) to make four main statements: (1) the LPI represents the stationarity of the system accurately; (2) the LPI is affected by population time-series with few records; (3) the LPI is sensitive to the initial decline of a few populations; and (4) the way zeros are handled affects the LPI. The authors use these four results to support a conclusion that "population time series used in the Living Planet Database cannot provide a reliable picture of current biodiversity changes" (lines 24-25). Overall, while this submission presents several technically correct results, I am of the view that it misinterprets these results to draw unsubstantiated conclusions.

We hope that the revised version does not include any unsubstantiated conclusions. Note that we need to balance the views of all the referees in the revised version, and the other referees do not share the concerns of the first referee, so we cannot be in complete agreement with the first referee in all cases. The second referee thinks that the LPI is untenable as a policy-relevant index and cannot be fixed. We do not want to be so strict, so we have attempted to take a careful position, not going beyond what can we show by our analyses.

My main issue is philosophical. It is not stated as plainly as this, but this submission implicitly presumes the existence of a normative indicator of global population trends, an empirical gold standard representing the 'true' state of populations. Any deviation from this unstated ideal is subsequently interpreted as a methodological flaw or a mathematical bias. To me, this is an informal logical fallacy known as the Nirvana fallacy (https://en.wikipedia.org/wiki/Nirvana_fallacy).

Indeed, it seems we are on a slightly different philosophical position. The LPI is a universally used index widely publicized in the media as a general indicator of the state of the biodiversity. We do not presume that any such indicator exists, but if we have an index which pretends to summarize overall changes of biodiversity in the last decades, we think it is important to reveal the flaws, if there are any. We do not say that a deviation from an ideal is a flaw (actually we claim the opposite), but we show the problems with the calculation of the widely used index. We certainly do not criticize the index for its inability to reflect the complex reality - it is the very nature of any index - but we show that the LPI does not represent correctly the „average“ trend (for which it has been constructed) due to several mathematical issues. We now clarify these issues in the Discussion section.

The LPI is an index - a quantitative summary of a much more complex natural phenomenon - in much the same way as other imperfect, yet still useful indices. The S&P500 is a composite index of US stock market. GDP is a composite index of a country's economic production. The Human Development Index is a composite index of a country's development status. We accept that each of these indices are useful without being perfect. Companies go bankrupt even when the S&P500 index rises, but we do not interpret this as evidence that the index itself is mathematically flawed or biased.

We feel there is a misunderstanding. Yes, of course, an index is not incorrect if it does not comprise everything and even if it provides misleading guidance in particular cases. But the way how it is compressing the reality should be mathematically correct and unbiased, which is not the case of the current calculation of the LPI. If there are mathematical problems in the calculation of the GDP or Human Development Index, they should be criticized, and eventually corrected, regardless of its ability or inability to reflect all the intricacies of reality. It is OK if an index provides a quantitative summary of a complex natural phenomenon, and we are far from criticizing this aspect, even though we agree that some of our statements in the previous version of the Discussion section could imply this (and we hope now we avoid such statements). But we show that such a quantitative summary could be flawed exactly in this respect, and there are ways how to correct and improve it.

I believe the authors make a fundamental mistake when they dismiss the analytical process of the LPI as 'arbitrary' (line 250). Some steps in the analysis may be 'subjective', but this is not the same as being arbitrary. The choices around including short time-series, weighting taxa and regions, averaging across realms and systems, and handling of zeros were not taken at random, but were judgements based on analytical trade-offs. Making different analytical decisions does not universally improve the LPI, it simply introduces a different set of trade-offs (which I describe below).

We agree that the term „arbitrary“ was not appropriate, and now we have changed it to „subjective“, as suggested. We absolutely agree that making different analytical decisions does not necessarily improve the index, and we do not criticize that these decisions have

been made. The problem is, however, that, according to our analyses, all these decisions have led to considerably a more decreasing index in comparison to other potential decisions that would be equally well justified. It is thus of the highest importance to evaluate the sensitivity of the index to these subjective decisions. We did it, and additionally we found some errors and decisions which are not well justified. We feel this is very important for a proper use of the index. We stress now that some subjective decisions must be done, but at the same time it is good to know what their consequences are.

The authors end their manuscript with the sentence "Relying on the LPI, even after the modifications we propose, provides incomplete information about the state of global biodiversity" (lines 335-337). This is a trivially true statement, and one that applies to any indicator of global biodiversity.

Indeed, we agree that this is trivial, and it did not follow from our analyses, so we have removed this sentence.

Similarly, they write (lines 76-77), "We also point out that the major issues related to the LPI are not only caused by the calculation itself, but are deeply related to the quality and representativeness of the underlying data". Again, this is trivially true for any dataset that is not a representative random sample of the population it is drawn from (i.e., almost every dataset of global diversity).

This is true, but we show that the issues directly related to the data set (extremely sparse data records, low number of time series at the beginning, unclear information value of zeros) are indeed responsible for the problems. We hope we are now clear in this respect.

In the first sentence of this review report, I purposely refer to this study as a 'sensitivity analysis' of the LPI because, to me, that is all that it is. It explores how the LPI responds to incremental adjustments to its calculation. This is a noble effort, for which the authors should be commended, but it is distinct from proving that the LPI is flawed or invalid.

As mentioned above, we do provide a sensitivity analysis (which we consider highly needed), but in addition to it, we point to several problems with the LPI which go beyond the simple issue of the sensitivity to subjective decisions. Anyway, we have reformulated the text to be clear in this respect. We think that if an index is very sensitive to individual settings, it should reflect it, instead of using just a narrow range of these possibilities.

What follows are specific comments on the four main findings of this study.

1. The LPI properly reflects the stationarity of the system

This section is technically correct. If a population is stationary and fluctuates randomly around that stable equilibrium, then the LPI is unbiased. By contrast, when populations fluctuate on a random walk, then the LPI is downward biased because individual populations diverge from the average response. This raises the obvious question of how populations are more likely to fluctuate, a question that has no readily available answer. I assume that sampling error would likely fluctuate around a stationary value (because an individual missed in a survey one year can still be recorded in the next); but population stochasticity would likely cumulate through time in a non-stationary way (because random births/deaths affect population dynamics in subsequent years).

Yes, exactly. We show that stationarity of fluctuations plays an important role on top of the symmetry on arithmetic/logarithmic scale - only the combination of arithmetic fluctuations and non-stationarity leads to the biased LPI. We believe that this is a novel point to the discussion about the nature of population fluctuations and the LPI. We have added a new figure with new simulations and more discussion on this issue, as also another referee asked for some extensions in this respect.

In this context, the authors may be interest in the comment (Talis & Lynch, 2023 <https://doi.org/10.1038/s41559-023-02086-w>) and response (Buschke et al. 2023 <https://doi.org/10.1038/s41559-023-02087-9>) published earlier this year. What matters to the LPI is not the relative frequency of different types of randomness (i.e. sampling error, environmental stochasticity, demographic stochasticity), but rather the magnitude of the form of randomness that does affect the index (i.e. demographic stochasticity)

We agree with the point of Buschke et al. that when arithmetic fluctuations are present, the LPI declines. However, this is only true for non-stationary arithmetic fluctuations (as modelled by the equation 1 in the paper). We show that if the arithmetic fluctuations are stationary, they do not lead to the decline of the LPI. This is a practically important point, because the presumably large and common arithmetic sampling errors (that are stationary)

would not contribute to the decline of the LPI. We have now incorporated both the references into our argumentation and strengthened it overall.

2. Number of records in the time-series

The authors are plainly correct that the LPI will be different if one removes time-series with few records. What is missing, in my opinion, is an assessment of how removing these shorter time-series affects taxonomic and geographic representativeness. I worry that there is a risk that using only high-quality time-series would turn the LPI into an index of birds from USA and Europe, possible in protected areas (which are more likely to be monitored over long periods, but less likely to show declines).

Yes, we entirely agree. The problem of the representativeness is indeed crucial, and this is the reason we pointed to it. Although removing population time series with low number of records reduces taxonomic and geographic representativeness, including them in the calculation introduces a much more severe bias due to sampling stochasticity that causes a decrease in the index. Note that about 20% of all time series contain only two data points. Removing such time series indeed increases the geographic bias, but including them generates even more substantial bias due to the issues now explained in the newly added figure.

I checked the 2022 public version of the Living Planet Database and saw that the average number of records per time-series Africa (7.4) and Latin America (6.7) is much less than Europe (14.6) and North America (18.6). Similarly, birds (14.9) tend to be surveyed for longer than amphibians (5.7) and even mammals (7.6).

I am not claiming that removing short time-series is necessarily wrong and that the LPI is right, but rather am illustrating the point that addressing one bias in the LPI (removing short time-series) can inadvertently introduce a different bias (taxonomic and geographical).

We agree with the referee that both the solutions have their pros and cons and it is necessary to deal with this trade-off between the reasonable number of records in time series and geographic bias. The problem we point out is that time series with low number of records are typically less reliable and, as we now show in the new figure, they may generate significant bias, reflected by the decreasing LPI. The removal of these population time series is thus a better solution. We have also added a new text addressing these issues.

An alternative argument - and one that I personally support - is that these results should not justify the exclusion of short time-series in the calculation of the LPI, but should rather stimulate the collection of better quality data for those populations with inadequate sampling. My view is that unintuitive results should encourage action to improve data, not disregard what limited data we already have. (I acknowledge that my interpretation is subjective, but it is an equally valid interpretation of the authors' results).

We have now reformulated the Discussion section in this direction. Unfortunately, we can hardly obtain better data for the beginning of the study period, so that we simultaneously need to make the index as unbiased as possible.

3. Extreme sensitivity of the LPI to the initial decline of a few populations

This result has been presented before by other studies (by Brian Leung and those commenting on his paper), so I have no reason to doubt it. However, it presents another example of, what in my view, is only one possible solution to a real problem. If the LPI is sensitive to a few anomalous trends from poorly studied taxa from understudied parts of the globe, the onus should be on collecting better data, not removing what little data there is. To me, removing messy data from poorly studied regions and taxa makes the LPI less representative of global biodiversity, not more.

We are not sure we agree. The sensitivity of the LPI to a few non-representative time series at the beginning of the study period is a real problem, as it compromises the generality of the index. Including these time series does not ensure any increase of the representativeness if they do not represent any reliable sample of the regions and taxa for a particular period. Note that Brian Leung and colleagues have shown something slightly different, namely that a few extremes of rapidly decreasing populations affect the average trend. Our findings agree with it, but we additionally show that the index is sensitive to the small sample size at the beginning of the study period, as the LPI is cumulative, so that if there is a decrease at the very beginning (even if it is due to the poorly representative series), the index can hardly increase later on even if its representativeness greatly increases towards the present. We hope we are now clear in this respect.

The caption to Figure 2 implies that there were three taxa, only one of which (*Vipera berus*) was declining. If this is the case, then the question can be posed to the authors why they are confident that the two remaining taxa are more representative of amphibians and reptiles in the Palearctic? (i.e., how can we be sure that these taxa are not resilient species that are more likely to be studied?)

This was a misunderstanding, and we are sorry for the confusion. The three taxa were (1) birds, (2) mammals, and (3) herptiles, and herptiles were represented just by one population for the period 1977-1978. The problem is that the two remaining taxa (birds and mammals) were well represented by many populations in that period, but since herptiles were represented only by this one population, the averaging has led to extreme overrepresentation of that one declining population (in fact, just four population records). In other words, an index which pretends to represent changes of thousands of studied populations, in fact strongly depends on just one population in this case, since at the beginning of the study period, no other population time series of herptiles existed (in contrast to many populations of other taxa). The cumulative nature of the index (the value of each year is calculated based on the value of the previous year) then affected the overall index value. We agree that our caption to Figure 2 was misleading in this respect, and we have clarified it.

But there may also be a technical point of confusion here, because I only have access to the public Living Planet Database from 2022 and it includes 54 time-series from 25 species of amphibians and reptiles in the Palearctic. In this version of the data, *Vipera berus* has data from six different populations (from three countries). The one population from Sweden, which declined between 1974 and 1981 is marked in the metadata as having been excluded from the final calculations (I am unsure why, it seems to be an island population, so highly variable population sizes seem realistic). Nevertheless, it seems that this section (lines 157-172) is based on an outdated version of the Living Planet Database and might no longer be supported by empirical evidence.

The problem is that the index values are calculated cumulatively, so that the mean growth rate for each pair of consecutive years is calculated separately, and the value for given year was calculated using this growth rate and the value of the index for the previous year. There are indeed 54 time series of herptiles for the Palearctic, but for years 1974-1977 there is only this one population of a viper in Sweden, so that the index values for these particular years are determined only by this single population. It clearly follows from our comparison of the LPI with and without this single population - if the other populations of herptiles outweighed this effect, the LPI would be similar in both cases. Note that an update of the dataset does not resolve the problem - we may remove one population of viper we used for our illustration, but there will be such single-population representatives in any subset of data used for particular LPI calculations. We have added a discussion on this issue.

4. The problem with zeros in the population time-series.

The authors highlight an important issue about zeroes, but I wonder if they could have made more of an effort to understand the zeroes in the Living Planet Database. I took a quick look at the public version of the database from 2022 and found that of the populations with zero values, 66% were birds, 24% were ray-finned fish, and 8% were mammals. The survey methods for fish were mostly catch data, where one could assume that low catch rates are due to reduced population densities or at least changing size frequencies that are less likely to be caught in larger net sizes (in which case, removing zeros would downplay these population reductions/changes).

We completely agree that while some zeros just reflect sampling issues, other zeros are ecologically informative, as they represent real extinctions. But our point is different. We argue that even if the zeros are ecologically informative, we cannot treat them by the same way as the non-zero values, because the calculation of the LPI is (correctly) based on the multiplicative population growth and thus cannot include zeros. And, importantly, any way to replace zeros by a (small) non-zero values seriously distorts the calculated population growth. It is thus not only a problem of the reality of the zeros (i.e. whether they reflect extinctions, emigrations, or unsampled populations due to low population density), but rather a mathematical problem that lies in the core of the calculation of the index that reflects multiplicative population growth. Even if we were able to distinguish ecologically informative zeros from the sampling issues, we would not be able to treat them mathematically consistently. For this reason, we strongly recommend to deal with the zeros separately from dealing with multiplicative population changes, i.e. to take the dynamics of colonization and extinction as a separate issue, which is certainly not less important than multiplicative population changes, but simply different. Note that this issue does not concern only the LPI but essentially all studies that focus on population dynamics and need to deal with both population fluctuations (the realm of classical population biology) and colonization-extinction dynamics (the realm of metapopulation biology).

Equally interesting is that 78% of the bird populations containing zeros seem to come from the Australian Shorebird Monitoring programme. I am not familiar with this program, but it seems to be a citizen science program that records shored birds twice a year. It is unlikely that these zeros represent extirpations and recolonisations, but rather migratory species that either were missed in surveys (presumably due to lower densities) or did not appear during the survey period (presumably due to the vagaries of shorebird migration). In both instances, it could be argued that these zero values are ecologically informative. This raises an issue not considered by the authors about whether the LPI should only consider closed populations (i.e. those unaffected by migration dynamics).

We agree. But again, the issue whether the zeros are informative and reflect extinctions in closed populations, emigrations in open populations, or sampling artifacts, is different, and although it is important, it is difficult to resolve by the data we have. Our concern goes deeper into the mathematics beyond the LPI and is more general.

In summary, I think this is another example where the authors are too quick to advocate for a specific analytical choice, instead of reflecting on whether exploring the imperfect LPI can yield deeper ecological insights about the state of biodiversity.

We are sorry that the referee considers our approach as too quick. We are aware about all the different pipelines used for the calculations and all the decisions which had to be made, as well as about various problems related to the data quality. We do not advocate for specific analytical choices but show that different choices have different consequences, and point to a few objective mathematical problems, with the aim to improve the LPI calculation.

To conclude, I am of the view that the authors draw the wrong conclusions from their otherwise technically correct analyses. The shortcomings of the LPI and the Living Planet Database should not discourage their use, but should rather stimulate action to improve these resources.

We agree, and this is actually what we already put into the Discussion section. However, we point to some problems which, as stressed by the second referee, seem to be really substantial and cannot be fully fixed by better data. We propose some ways how to improve the calculation of the LPI; this has been exactly the purpose of the study.

Finally, I am certain the authors agree with me (and the originators of the LPI) that global vertebrate populations cannot be summarised in a single metric. Instead, the value of the LPI is to provoke action to collect better data or explore nuanced analyses. This is demonstrated through the consistent growth in the number of population time-series in the Living Planet Database over the last two decades. To preempt any potential concerns that summarising the LPI in a single metric can create the wrong public impression, I would like to point out that the 2022 version of the Living Planet Report was also covered in the media in a very different way. High profile pieces in the New York Times (<https://www.nytimes.com/2022/10/12/climate/living-planet-index-wildlife-declines.html>), Vox (<https://www.vox.com/down-to-earth/2022/10/12/23399105/biodiversity-loss-wwf-living-planet-index>), and Smithsonian Magazine (<https://www.smithsonianmag.com/smart-news/animal-populations-faced-a-very-sharp-decline-since-1970-180980957/>) made a concerted effort to explain the nuance of the LPI and how it might be interpreted.

Indeed, the situation in presenting the LPI is improving, and we are grateful for that. Unfortunately, it is still very common to find in various media very inaccurate description of the LPI leading to confusing misinterpretation. Importantly, the issues we raise in our study have not been yet appreciated.

I think this is a very valuable development, and it shows the usefulness of having a single metric as a gateway to deeper reflection and public education.

As mentioned above, we did not criticize the use of a single metric for showing average trends; this is a common practice and people know that even if the world is complex, we need to simplify it sometimes. But the problem is that there are serious issues with this particular index, which need to be addressed. We provide a guidance how to address these issues.

Reviewer #2 (Remarks to the Author):

This ms takes by far the deepest dive into the LPI to date and finds some pretty substantial problems.

We agree that the problems we raise are substantial.

First, LPI is probably the single most widely known and used index of human impact on biodiversity available. As noted in the first paragraph this has gone from an index that receives exceptionally wide press coverage to one that is embedded into UN conventions and treaties. Thus understanding its strengths and weaknesses is critical and entirely NatureComm appropriate.

Thank you!

At the same time my read of the conservation community is that they tend to not trust the LPI and see it as badly exaggerated without knowing why. Several recent papers have hinted at why there might be problems. This paper reviews prior efforts and then takes a much deeper dive to arrive at strong answers. Several methodological issues are discussed (e.g. zeros) but the real problems appear to lie in the nature of the data (groupings and weightings with sparse coverage, zeros in the data, etc).

We pointed this issue in the original version of the manuscript, and we agree with it, but since the first referee was not happy with stressing it out, we have slightly toned down this part.

To be honest my conclusion after reading this paper is that the LPI is untenable as a policy-relevant index and cannot be fixed. Others might not draw quite as strong a conclusion, but this debate needs to be had.

Personally, we would agree, but the opinions on this issue seem to differ in the community (see comments of the reviewer #1). We structured the new version of the manuscript so that it provides description of the biases in the LPI calculations, but leaves the final opinion on the readers. We agree that the debate is needed, and our aim was to contribute to this debate.

The paper is clear and scientifically correct. I have relatively little suggestion for improvement.

My one suggestion is that the finding that length of time series (by # of records) is important with short time-series being biased to decreases, which has been noticed more glancingly by a few other papers (e.g. Leung 2019 and the commentaries). I did not see a clear explanation (either conceptual or data) why the short time series are biased down - it is not obvious that this has to be. Anything the authors can add to this debate would help make this a "one-stop" paper on the LPI critiques.

This is a very good point. We provide some explanations in the new figure as well as in the text.

Reviewer #3 (Remarks to the Author):

This manuscript focusses on the LPI for biodiversity trends estimated for vertebrates from a large collection of (biologically and statistically heterogeneous) time series. The authors raise some concerns about data quality that are echoed elsewhere, but they also make some points about the computation methods and assumptions that go into estimating the index.

Overall, I think the manuscript makes some good points, but I feel some of these points are more general than to the LPI, and this be made more clearly than in the current version of the text.

We agree, and we clarified that some of the points really have more general relevance.

For example, potential biases caused by not including measurement error, immigration/emigration processes would affect any method of trend estimation, and not just the LPI. There is a danger that by focussing very narrowly on the LPI, some of these wider points are not absorbed by the more general reader. My suggestion would be to try and widen the scope of the points raised, to go beyond the LPI.

This is not as easy, as our study is focused on the LPI and its length is restricted, so we cannot be too broad. We think that the focus on the LPI is justifiable due to its wide use, so we have felt it is important to deal with the issues specific to the LPI. However, we tried to stress the broader implications of our findings where possible.

Early on it is pointed out that different methods lead to different conclusions regarding biodiversity trends (refs 8 and 9 are used here). However, it is fair to say there will be methodological biases in these approaches too because they use simplifying assumptions. For example, ref (9) does not appear to

include process error in their trend estimates, nor is it clear any model validation is done. My point is that some of the issues raised in this manuscript are more general than to the LPI.

We agree. But unlike the other approaches, the LPI is so commonly used as a singular result in public debate and decision making that it deserves special scrutiny in our opinion. We now mention that there are simplifying assumptions also in the other approaches we mention, and point out that exploring sensitivity to ad hoc assumption is a recommended practice in all similar approaches.

The manuscript results section begins with the rebuttal of some recent criticisms of the LPI analyses regarding the estimation of mean trends. I very much agree with the points raised by the authors of the current manuscript, but I do feel that this section should be supported by some analyses (figures and, where appropriate, maths and/or code). This would strengthen the argument in lines 86-87: "We found out that the problem with this simulation strategy is that it leads to unrealistic non-stationary population size distributions."

We agree that this is important, and we expand our arguments in this respect (see the new figure and new discussion in the text).

The effect of the number of records in the time series. Whilst I agree with many of the points made here, I feel there are other considerations. A key question for me is also whether short time series are expected to be biased? I expect them to be noisy, but, if we simulate a large number of trends with the same underlying trend, do we expect a bias from the averaging of these short times series? The data-poor time series might be declining because there is a bias towards monitoring declining species, but this is a bias created by the non-random species/population sampling rather than a bias from short (data-poor) time series per se. It is good to point out how including/excluding the data poor time series changes the LPI, but it should open more of a discussion about what is going on in these populations, and how we should treat the data associated with them. It isn't clear that a one size fits all approach is the best (ie including or excluding all data below a certain length).

This is a very good point, similar to a point raised also by the second referee. We now provide some explanation of these effects. One reason is that data-poor time series are processed by the chain method which preserves stochastic noise (which is typically quite high in data-poor time series) at the arithmetic scale, leading to the declining index, while data-rich time series are processed by GAM which does not have this effect. We discuss these issues, and provide some statistics concerning short time series.

I would also suggest the data poor time series could be weighted. Indeed this is what has been done in this manuscript -short time series are effectively given a weighing of zero in supplementary figure 1 etc. But there might be other useful ways to weight time series that include as much of the data as possible, but acknowledges uncertainty in the individual trends based on the number of sampled time points. This is similar to a meta-analysis approach, and would very likely place the estimated biodiversity trends in between the LPI and adjusted-LPI.

This is an interesting idea. Weighting the time series based on their information value would be a good remedy to the sensitivity of the LPI to the idiosyncratic population stochasticity. On the other hand it would not address the above mentioned problem of the chain method and sampling bias. We now mention this possibility in the discussion.

Like many of the issues raised in this manuscript, these are decisions all methods need to consider when estimating general trends/indices of biodiversity change.

Indeed - as pointed also by the first referee, some decisions had to be made. And we do not criticize the fact that the authors of the index made decisions, but we point out that some of the decisions have led to extreme sensitivity of the index to data-poor and unreliable time series, as well as to the bias towards the declining LPI.

Extreme sensitivity of the LPI to the initial decline of a few populations
This section raises similar issues to the last one, and I think the authors make a good point about taxa being represented by just a few populations this point should also apply to the previous section.

We agree.

The problem of zeros in population time series
This section discusses how biases may arise in the LPI calculation because the growth rates are log-

transformed, and when there is a 0 this will cause well-known problems. As pointed out the LPI uses one of several options. I think the other options should be made explicit.

We have reformulated this part, as it was a bit confusing. In fact, there are only two solution, one which is used by the LPI calculation (i.e. replacement of zeros by another value, which necessarily distorts the resulting growth rates, as we explain in the text), the other is suggested by us, i.e. complete removal of zeros.

In my view this problem highlights a wider problem - the populations are generally not closed. Any time point that has a 0 and is followed by a non-zero number shows that either (i) there is observation/measurement error; or (ii) there is immigration/emigration in this system. I don't think the authors consider observation error in their arguments, but it could explain many 0's. However, if there is immigration/emigration then other time points will be biased too, assuming the model does not take these into account. Even without the $\log(0)$ problem, trends could be biased if they do not take these processes into account, and so the problem goes beyond considering only the 0 time points ie dealing with zeros in a different way does not get round the point raised about immigration/emigration processes.

We completely agree. This really goes beyond the LPI, and it is a bit different issue than the mathematical problem we point out. We mention these related problems in the Discussion section.

Finally, I agree that the coding errors reported are important to have catalogued so they can hopefully be addressed if the LPI is used further down the line.

Thanks!

Reviewer #1 (Remarks to the Author):

(I was Reviewer 1 in the original submission)

After reading the revised manuscript and the authors' respond to previous review comments, I am unconvinced that the authors have incorporated my earlier comments into a revised manuscript. Therefore, my original assessment remains unchanged.

I have also seen comments by the other reviewers, so I understand why the authors chose to focus on those more complimentary reviews than on my concerns. Peer-review does not have to be unanimous, and I understand that handling editors have to evaluate the relative merits of conflicting review reports when making judgments. So, I do not feel it is useful to repeat all my concerns here. I will, however, re-raise two points:

First, I cannot support using the example of the viper in Figure 4. To me, it seems disingenuous to choose a time-series from a population that is explicitly excluded from the Living Planet Database (lines 185 – 191). There are more than 100 such exclusions out of roughly 32,000 time-series in the Database, so the authors are purposely picking a time-series with a recognised quantitative artefact to make their point. While I acknowledge that they try to explain this in line 185 – 191, I am of the opinion that this is insufficient context relative to Figure 4 and, again, the description in the discussion (lines 282 – 285).

Related to this, the authors continue using an outdated version of the Living Planet Database (from January 2022), even though they have been made aware of a much more recent version (the latest version has more than 10,000 extra populations records and included about 200 extra species). While I don't contend that this updated data will necessarily change the main results, do not think the authors should be using a dataset that is more than 2 year out of date.

Second, I do not believe that the authors made sufficient effort to evaluate the consequences of their recommendations, despite being provided with quantitative summaries during the earlier round of review. I appreciate that they added descriptive supplementary Figures, but the significance of these data are downplayed in the main manuscript. For example, Figure S4 makes a strong case that limiting the LPI to time-series with more than five records will disproportionately favour data from Europe and North America, yet in line 150 the authors' claim that "the problem of representativeness is accounted [for] by weighting and thus is not as serious". Missing from Figure S4 is data on the total number of species for each combination of system-realm-class because the figure shows percentages. Essentially, removing shorter time-series will remove data from the combinations of region and class that carry the highest weight in the index, which, to me, is a major point of concern (the authors seem to agree with my concern because in line 334 when they write that weighting "introduces another bias as the weighted index is driven mainly by tropical populations (species-rich regions)").

Related, there was little effort to diagnose the prevalence of zeros in the Living Planet Database, simply claiming in line 257 that "...the methodology of the LPI calculation is unable to properly treat them". The issue is only partly about the LPI calculation, but more about whether removing zeros is a better or worse choice than adding 1% of the mean population. If the zeros are ecologically meaningful (rather than sampling artefacts), then removing zeros will bias the index upwards (because downward dips, even if temporary, will not be considered by the calculation).

Specific comments

Figure 2 and 3: I realise that these additional figures were in response to request from other reviewers, but I do not see how they add to the manuscript (or what is already known in the literature; and cited by this submission). Perhaps these would be better suited for the Supplement?

Lines 15: If the authors are concerned about the LPI being misinterpreted, then they should use the correct definition of the index in the abstract. The LPI is an indicator of the average change in the relative abundance (or proxies of abundance) of monitored populations of wild vertebrates (i.e., it is not the overall population trend).

Line 39: This is outdated. The LPI is not a headline indicator for any of the Goals or Targets of the GBF (See CBD decision CBD/COP/DEC/15/5). It is still a component and complementary indicator. Please update reference 6 because the zero draft has been superseded by the Framework itself (CBD/COP/DEC/15/4).

Lines 186-190: This newly added section would benefit from an extra round of proof-reading for clarity.

Line 258: Perhaps the issue is not how the zeros are handled, but rather the choice of model use to estimate trends. See: O'Hara & Kotze (2010) Do not log-transform count data. *Methods in Ecology and Evolution*, 1, 118-122.

Lines 305 – 307: I don't understand this additional sentence. What is meant by 'cannot' outweigh the problem of short time series. Do the authors mean that the bias 'should not' outweigh, or that the bias 'is unable' to outweigh the negative effect of short time-series. The former makes sense and is defensible, but the latter needs further justification before it should be accepted.

(Remarks on code availability):

Since I believe the submission has major scientific shortcomings, I didn't feel it was necessary to review the code because it wouldn't change my assessment.

Reviewer #2 (Remarks to the Author):

Thank you for the opportunity to re-review this mss. My impression remains the same - this is an important contribution about a highly influential index. One cannot have an index becoming increasingly central to policy that is immune to criticism and analysis.

Upon this 2nd read, I especially noticed the detailed summary of the exact calculation method (sequence in which weightings are applied etc). This is a more thorough explanation than any that I have found from the LPI folks which mostly cite papers that are quite old and incomplete. This alone represents a valuable contribution.

One minor point. There were several grammar errors. Insertions of extra "an" and "how" were what I noticed most often. It did not distract from the reading and I am sure can be fixed in copy editing. However on line 278 (page 10), I believe there is a word choice that gives the opposite meaning. I believe it should read "This problem is AGGRAVATED by the specificities of the data,..." (current word is ALLEVIATED which means lessened or made better, where as AGGRAVATED means increased or made worse).

Overall I found the claims strongly supported by evidence, and the discussion balanced and fair. If anything the discussion section per se seems a bit watered down compared to the findings reported earlier.

As requested I share my thoughts on the other reviewers feedback.

My sense is that reviewer #3's comments were mostly addressed in the revision, except for the recurring point that many of the issues raised exist in any effort to make a global index. While the last point is true, I am sympathetic to the authors view that it is a different paper to discuss general qualities of good indices and how to construct good indices, and also that the LPI stands so far above other efforts in usage/recognition/impact that it is fair to single it out for analysis, even necessary for the sake of good policy analysis.

Reviewer #1 re-review

1) The use of the viper example. I am unsure what to recommend here. If the authors maintained

the viper example because it was too much work to find another equally good example, I would agree with reviewer #1. If the authors maintained the viper example because it was the best example/only example quite that extreme which illustrates the more general point of significant impacts of single populations in the subgroupings (e.g. in all other cases it was two populations early in the 1970-today range), then I am sympathetic with the authors. I cannot tell which case applies from the author rebuttals. I don't agree with reviewer #1 that it is beyond the pale to use the viper example just because it was excluded in 2024 - it was as best I can tell included in many earlier published instances of the LPI (which has been published every two years for a decade or so) and it is illustrative.

2) I don't see any point in making the authors redo their analysis with hot off the press data. That is a lot of work, and even reviewer #1 acknowledges it won't change the results. Publication is not instantaneous and can reasonably use data that was available at the time research starts unless there is an awareness that errors were identified and corrected in newer data, making it a waste of time to critique old data. This does not apply here. Notably when I go to get LPI data, I still see the 2022 as what is being primarily served to the public (I haven't spent enough time to say the 2024 data is not available to the public but it is not what is presented front and center by the LPI at the time I write this e.g. the headline page "Latest Results" goes to 2022

https://www.livingplanetindex.org/latest_results).

3) In contradiction to reviewer #1, I feel like the authors do adequately acknowledge that the removal of short time series mostly removes data in a geographically biased fashion, but they do this to counteract a bias and note (correctly) that weighting is at least an unbiased correction, whereas including known biased data is a biased (statistically speaking) choice. Bottom line I am much more sympathetic to the authors line of argument (no known bias) than reviewer #1 (continue to incorporate a bias because more data is always better).

4) At a minimum, I found figure 3 added value - it shows visually the impacts of using the geometric average amidst arithmetic sampling noise - a key issue with the LPI that is hinted at but not fully fleshed out anywhere to my knowledge. And as one of the values of this paper is a "one stop" list of issues with LPI, I favor keeping it. Figure 2 I am more ambivalent about. I would be fine with it going into supplemental. The debate about how the Buschke paper was wrong is a bit technical and a bit of a distraction in my mind.

(Remarks on code availability):

Code was primarily LPI code which I have read at other times. The point of this paper was a critique of LPI code/methods/choices.

REVIEWERS' COMMENTS

Reviewer #1 (Remarks to the Author):

(I was Reviewer 1 in the original submission)

After reading the revised manuscript and the authors' respond to previous review comments, I am unconvinced that the authors have incorporated my earlier comments into a revised manuscript. Therefore, my original assessment remains unchanged.

I have also seen comments by the other reviewers, so I understand why the authors chose to focus on those more complimentary reviews than on my concerns.

We have done our best to address your concerns in the current version, while having in mind also comments from other reviewers.

Peer-review does not have to be unanimous, and I understand that handling editors have to evaluate the relative merits of conflicting review reports when making judgments. So, I do not feel it is useful to repeat all my concerns here. I will, however, re-raise two points:

First, I cannot support using the example of the viper in Figure 4. To me, it seems disingenuous to choose a time-series from a population that is explicitly excluded from the Living Planet Database (lines 185 – 191). There are more than 100 such exclusions out of roughly 32,000 time-series in the Database, so the authors are purposely picking a time-series with a recognised quantitative artefact to make their point. While I acknowledge that they try to explain this in line 185 – 191, I am of the opinion that this is insufficient context relative to Figure 4 and, again, the description in the discussion (lines 282 – 285).

We used the example of the viper to illustrate the sensitivity of the LPI calculation to sparse data at the beginning of the study period. This problem will be persistent regardless of this particular population because any newly added population time series in an updated version of the Living Planet Database (LPD) will most likely not extend back to the beginning of the study period. Therefore, the LPI will still fully depend on the few population time series available at the beginning of the study period.

According to the information in the current version of the LPD, the viper population (as well as some other populations) is not included in the latest LPI calculation. However, it does not remove the problem. Even after excluding populations marked as excluded in the LPD and the replicates which were not used in the LPI, there are still single-population representatives or 2/3/4-population representatives (whose effect is similar to single-population representatives) for an entire taxon. For example, Palearctic herptiles are represented again by a single population, this time by a skink from Japan, in 1978 and 1979 (the skink population became the only representative of Palearctic herptiles after excluding the viper). Neotropical birds are represented by only two populations for the first years of the study period, and then only by three populations (note that this taxon and realm have a particularly high weight in the calculation). And there are more examples of underrepresented taxa. Additionally, many of these populations consist of only two records, which leads to considerable sampling-based bias (for example, one of two representatives of freshwater Neotropical herptiles, followed by four of five representatives, with the second and last record being 0 for all, which introduces another problem; or freshwater Indo-Pacific fish represented by one population; both examples have again a high weight in the calculation).

Therefore, we retained the viper example as we used the LPD2022 (not the same version as was used in the latest Living Planet Report 2022), but we have added some more explanation into the main text. The viper population serves as an illustrative example that is

still valid for the partial LPI (for example, for the Palearctic LPI as we show in Fig. 5, formerly Fig. 4) because the excluded populations concern only the global LPI. When calculating the LPI at a regional or national level or for a specific taxon/realm/habitat (which is quite common), taxa will often necessarily be represented by just one population, resulting in the problems we have described.

Related to this, the authors continue using an outdated version of the Living Planet Database (from January 2022), even though they have been made aware of a much more recent version (the latest version has more than 10,000 extra populations records and included about 200 extra species). While I don't contend that this updated data will necessarily change the main results, do not think the authors should be using a dataset that is more than 2 year out of date.

We have been working on this study for several years and always used the data which were available at the moment. The updated version of the Living Planet Database does not contradict our conclusions about the index nor change them in any way, see our discussion with referee #2 on this topic. Additionally, none of the LPD versions had all the data publicly available, so the index recalculations (by other authors than the LPI team) would never be the same as those reported in the Living Planet Report. Anyway, working with a subset of outdated data does not preclude evaluating the LPI methodology. All the shortcomings we pointed out in the manuscript would persist for any version of the LPD, even though they would obviously result in slightly different quantitative patterns/comparisons.

Second, I do not believe that the authors made sufficient effort to evaluate the consequences of their recommendations, despite being provided with quantitative summaries during the earlier round of review. I appreciate that they added descriptive supplementary Figures, but the significance of these data are downplayed in the main manuscript. For example, Figure S4 makes a strong case that limiting the LPI to time-series with more than five records will disproportionately favour data from Europe and North America, yet in line 150 the authors' claim that "the problem of representativeness is accounted [for] by weighting and thus is not as serious". Missing from Figure S4 is data on the total number of species for each combination of system-realm-class because the figure shows percentages. Essentially, removing shorter time-series will remove data from the combinations of region and class that carry the highest weight in the index, which, to me, is a major point of concern (the authors seem to agree with my concern because in line 334 when they write that weighting "introduces another bias as the weighted index is driven mainly by tropical populations (species-rich regions)").

We agree, but these arguments do not contradict our conclusion. Tropical regions are indeed represented by a smaller number of populations compared to temperate regions. This is compensated by weighting by species richness of specific taxa and biogeographical realms (their relative proportions). The global LPI is thus driven mainly by species-rich but data-poor tropical regions. The problem is that these tropical population time series can be more biased than the temperate data, as tropical populations were more often collected only in specific localities and for specific purposes.

It is true that excluding populations with e.g. less than five records reduces the representativeness in terms of the number of processed populations. However, we feel it is not appropriate to include them just to increase representativeness in a situation when including them leads to bias described in Fig. 4 (formerly Fig. 3). An extreme case are populations with two records, where it is indisputable that even if all of them were stable, the sampling error causes a decreasing index (see Fig. 4). The reason is that sampling-based deviations from the true population value are symmetric on the arithmetic scale, but the LPI calculation is based on geometric averaging of population changes, so the resulting index is necessarily lower than

1 (Fig. 4). We can discuss how many record-poor time series to include, but the LPI should not be based on the sampling-driven bias.

We have added the table with the total numbers of populations for each combination of system-realm-class, as suggested by the referee.

Related, there was little effort to diagnose the prevalence of zeros in the Living Planet Database, simply claiming in line 257 that "...the methodology of the LPI calculation is unable to properly treat them". The issue is only partly about the LPI calculation, but more about whether removing zeros is a better or worse choice than adding 1% of the mean population. If the zeros are ecologically meaningful (rather than sampling artefacts), then removing zeros will bias the index upwards (because downward dips, even if temporary, will not be considered by the calculation).

We disagree. Our point is not that we cannot distinguish zeros that are ecologically meaningful from those caused just by sampling errors (and leave only those meaningful). Even if we were able to distinguish and utilize only time series with ecologically meaningful zeros, the problem would remain. Population fluctuations that include values replacing zeros (values two orders of magnitude smaller than the mean population size), deliberately distort the statistical properties of the population trend. The issue is thus not whether this arbitrary population change is too large or not. When we have a procedure that deals with population growth (log-transformed data and ratios), we can utilize data only with non-zero population values.

It is appropriate to deal with zeros by an additional analysis (e.g. to separately analyze rates of colonization and extinction), or use methods that are suitable for dealing with time series with zeros (e.g. using some likelihood approaches), but these approaches are incompatible with the original population growth-based LPI methodology. We do not claim that the LPI methodology is inappropriate – on the contrary, it is a good tool when corrected for the biases pointed out by our study, but it is simply not suitable for dealing with zeros. We demonstrated the extent of the effect of zero values in the time series by removing zeros from the population time series and showing the resulting LPI.

Specific comments

Figure 2 and 3: I realise that these additional figures were in response to request from other reviewers, but I do not see how they add to the manuscript (or what is already known in the literature; and cited by this submission). Perhaps these would be better suited for the Supplement?

We consider these figures very important and we are very grateful to the referees that they forced us to make them – they considerably strengthen our results and in fact they provide new essential arguments why (and when) we should expect bias in the LPI calculation. We thus prefer to keep them in the manuscript.

Lines 15: If the authors are concerned about the LPI being misinterpreted, then they should use the correct definition of the index in the abstract. The LPI is an indicator of the average change in the relative abundance (or proxies of abundance) of monitored populations of wild vertebrates (i.e., it is not the overall population trend).

But the Living Planet Report also states that "the LPI results are calculations of average trends". We do not see a substantial difference between "average change in the relative abundance" and "overall population trend" (although we understand the gist of the difference). Anyway, we have reworded the Abstract according to this suggestion.

Line 39: This is outdated. The LPI is not a headline indicator for any of the Goals or Targets of the GBF (See CBD decision CBD/COP/DEC/15/5). It is still a component and complementary

indicator. Please update reference 6 because the zero draft has been superseded by the Framework itself (CBD/COP/DEC/15/4).

Thank you, we have corrected it.

Lines 186-190: This newly added section would benefit from an extra round of proof-reading for clarity.

Yes, we did it.

Line 258: Perhaps the issue is not how the zeros are handled, but rather the choice of model use to estimate trends. See: O'Hara & Kotze (2010) Do not log-transform count data. *Methods in Ecology and Evolution*, 1, 118-122.

We entirely agree that there are models that can deal with zeros. However, the LPI methodology is based on the geometric mean of relative population sizes. Using a different model, e.g. based on quasi-Poisson or negative binomial as in the above paper or in the recent paper we have already cited (Korner-Nievergelt et al. 2022), would mean completely readjusting the methodology - and that would be a completely different approach than the LPI. Although these models also have their pitfalls, adding them alongside the LPI can be one of the improvements to the calculation method, as we have already stated in the manuscript.

Lines 305 – 307: I don't understand this additional sentence. What is meant by 'cannot' outweigh the problem of short time series. Do the authors mean that the bias 'should not' outweigh, or that the bias 'is unable' to outweigh the negative effect of short time-series. The former makes sense and is defensible, but the latter needs further justification before it should be accepted.

We have reformulated the sentence to be clear. We have meant that reducing the number of time series after removing the sparse ones does not have as negative effect as leaving these sparse time series in the calculation, which bring a significant downward bias.

(Remarks on code availability):

Since I believe the submission has major scientific shortcomings, I didn't feel it was necessary to review the code because it wouldn't change my assessment.

Reviewer #2 (Remarks to the Author):

Thank you for the opportunity to re-review this mss. My impression remains the same - this is an important contribution about a highly influential index. One cannot have an index becoming increasingly central to policy that is immune to criticism and analysis.

Thank you.

Upon this 2nd read, I especially noticed the detailed summary of the exact calculation method (sequence in which weightings are applied etc). This is a more thorough explanation than any that I have found from the LPI folks which mostly cite papers that are quite old and incomplete. This alone represents a valuable contribution.

Yes, the intention was also to give a complete, precise and very detailed description of the LPI methodology.

One minor point. There were several grammar errors. Insertions of extra "an" and "how" were what I noticed most often. It did not distract from the reading and I am sure can be fixed in copy editing. However on line 278 (page 10), I believe there is a word choice that gives the opposite meaning. I believe it should read "This problem is AGGRAVATED by the specificities of the data,..." (current word is ALLEVIATED which means lessened or made better, where as AGGRAVATED means increased or made worse).

Thank you for the notice, we really made a mistake and actually meant "aggravated". We have rewritten it. We also got the manuscript proofread.

Overall I found the claims strongly supported by evidence, and the discussion balanced and fair. If anything the discussion section per se seems a bit watered down compared to the findings reported earlier.

As requested I share my thoughts on the other reviewers feedback.

My sense is that reviewer #3's comments were mostly addressed in the revision, except for the recurring point that many of the issues raised exist in any effort to make a global index. While the last point is true, I am sympathetic to the authors view that it is a different paper to discuss general qualities of good indices and how to construct good indices, and also that the LPI stands so far above other efforts in usage/recognition/impact that it is fair to single it out for analysis, even necessary for the sake of good policy analysis.

Thank you for your understanding. Yes, our manuscript is targeted at this particular index and even though there will certainly be many aspects common with other indices, a general discussion of them would distract the main message of the paper.

Reviewer #1 re-review

1) The use of the viper example. I am unsure what to recommend here. If the authors maintained the viper example because it was too much work to find another equally good example, I would agree with reviewer #1. If the authors maintained the viper example because it was the best example/only example quite that extreme which illustrates the more general point of significant impacts of single populations in the subgroupings (e.g. in all other cases it was two populations early in the 1970-today range), then I am sympathetic with the authors. I cannot tell which case applies from the author rebuttals. I don't agree with reviewer #1 that it is beyond the pale to use the viper example just because it was excluded in 2024 - it was as best I can tell included in many earlier published instances of the LPI (which has been published every two years for a decade or so) and it is illustrative.

We had thoroughly explored all the single-population representatives and their effects. The viper population is the only example of a single-population representative of an entire taxon (herptiles) for the whole region (terrestrial Palearctic) at the beginning of the study period (later, there are more representatives of the herptiles taxon). The other two taxa (birds and mammals) do not contain single-population representatives for the terrestrial ecosystem, although they sometimes contain just two- or three-population representatives. There are also single-population representatives for freshwater and marine ecosystems, described in the Supplementary Notes. We use the example with viper in the main text because it has the

strongest effect and well illustrates the extreme sensitivity of the LPI to single-population representatives especially at the beginning of the study period.

As we have replied to reviewer #1, we think there are several good reasons to keep the viper example in even though it was excluded in the most recent LPD. First, even in the updated LPD, there are still some single-population representatives later in the study period (or 2/3/4-population representatives). Their effect is currently weaker than the viper, but this may change with future exclusion/inclusion of data or change of the study period, and the conceptual issue remains. Second, although the viper population and several other populations were excluded for the global LPI, we do not have any indication whether they were excluded (and what was the exclusion criterion if they were) for the partial LPI calculations. Third, when calculating the LPI for some particular realm/class/habitat, different sparse population data at the beginning of the study period will necessarily appear. This situation would not improve with the updated versions of the LPD, since the newly added population time series would not typically overlap with the beginning of the study period.

2) I don't see any point in making the authors redo their analysis with hot off the press data. That is a lot of work, and even reviewer #1 acknowledges it won't change the results. Publication is not instantaneous and can reasonably use data that was available at the time research starts unless there is an awareness that errors were identified and corrected in newer data, making it a waste of time to critique old data. This does not apply here. Notably when I go to get LPI data, I still see the 2022 as what is being primarily served to the public (I haven't spent enough time to say the 2024 data is not available to the public but it is not what is presented front and center by the LPI at the time I write this e.g. the headline page "Latest Results" goes to 2022 https://www.livingplanetindex.org/latest_results).

The results of the LPI for 2020 and 2022 (the most recent one) are nearly identical. Our points would not change with the updated LPD, as the underlying principles remain the same (as we noted in the response to reviewer #1). Yes, we started the study with the 2022 dataset, which was a few months before its update used for the latest LPI2022, and probably other updates have already taken place (hard to judge from the attached material).

3) In contradiction to reviewer #1, I feel like the authors do adequately acknowledge that the removal of short time series mostly removes data in a geographically biased fashion, but they do this to counteract a bias and note (correctly) that weighting is at least an unbiased correction, whereas including known biased data is a biased (statistically speaking) choice. Bottom line I am much more sympathetic to the authors line of argument (no known bias) than reviewer #1 (continue to incorporate a bias because more data is always better).

Yes, incorporating the strong bias due to sparse time series is a much less appropriate solution than using only a subset of time series.

4) At a minimum, I found figure 3 added value - it shows visually the impacts of using the geometric average amidst arithmetic sampling noise - a key issue with the LPI that is hinted at but not fully fleshed out anywhere to my knowledge. And as one of the values of this paper is a "one stop" list of issues with LPI, I favor keeping it. Figure 2 I am more ambivalent about. I would be fine with it going into supplemental. The debate about how the Buschke paper was wrong is a bit technical and a bit of a distraction in my mind.

We would like to keep both the figures in the main text, as we find both of them very useful and important. Figure 3 (formerly Figure 2) is relevant not only to the debate on the Buschke paper (although this paper inspired it). It goes considerably deeper and explains when the LPI approach works correctly and why. It clarifies that the LPI necessarily decreases when there

is a divergence of population time series (the system is non-stationary) and population changes are symmetric on the arithmetic scale. And this happens in the case of sparse time series, as shown in Fig. 4 (formerly Fig. 3). Both the figures are thus essential and they complement each other, both of them corresponding to the respective main text.

(Remarks on code availability):

Code was primarily LPI code which I have read at other times. The point of this paper was a critique of LPI code/methods/choices.